# Anomalous magnetoresistance due to longitudinal spin fluctuations in a $J_{eff} = 1/2$ Mott semiconductor

Lin Hao [1,8], Zhentao Wang [1,8], Junyi Yang[1], D. Meyers[2], Joshua Sanchez[3], Gilberto Fabbris [4], Yongseong Choi [4], Jong-Woo Kim[4], Daniel Haskel[4], Philip J. Ryan[4,5], Kipton Barros[6], Jiun-Haw Chu[3], M.P.M. Dean[2], Cristian D. Batista [1,7] & Jian Liu [1]*

As a hallmark of electronic correlation, spin-charge interplay underlies many emergent phenomena in doped Mott insulators, such as high-temperature superconductivity, whereas the half-filled parent state is usually electronically frozen with an antiferromagnetic order that resists external control. We report on the observation of a positive magnetoresistance that probes the staggered susceptibility of a pseudospin-half square-lattice Mott insulator built as an artificial $SrIrO_3/SrTiO_3$ superlattice. Its size is particularly large in the high-temperature insulating paramagnetic phase near the Néel transition. This magnetoresistance originates from a collective charge response to the large longitudinal spin fluctuations under a linear coupling between the external magnetic field and the staggered magnetization enabled by strong spin-orbit interaction. Our results demonstrate a magnetic control of the binding energy of the fluctuating particle-hole pairs in the Slater-Mott crossover regime analogous to the Bardeen-Cooper-Schrieffer-to-Bose-Einstein condensation crossover of ultracold-superfluids.

[1] Department of Physics and Astronomy, University of Tennessee, Knoxville, TN 37996, USA. [2] Department of Condensed Matter Physics and Materials Science, Brookhaven National Laboratory, Upton, NY 11973, USA. [3] Department of Physics, University of Washington, Seattle, WA 98105, USA. [4] Advanced Photon Source, Argonne National Laboratory, Argonne, IL 60439, USA. [5] School of Physical Sciences, Dublin City University, Dublin 9, Ireland. [6] Theoretical Division and CNLS, Los Alamos National Laboratory, Los Alamos, New Mexico 87545, USA. [7] Quantum Condensed Matter Division and Shull-Wollan Center, Oak Ridge National Laboratory, Oak Ridge, TN 37831, USA. [8] These authors contributed equally: Lin Hao, Zhentao Wang. *email: jianliu@utk.edu

While a huge variety of unusual symmetry-breaking orderings can emerge as the ground state of correlated electrons, the disordered state above the phase transition is often even more enigmatic due to fluctuations that are challenging for experimental characterization and theoretical description[1]. This is particularly true when strong interplay between the spin and charge degrees of freedom is in play, such as the fascinating normal state of high-temperature superconductors[2]. One of the profound outcomes of the electronic spin-charge interplay is the Mott insulating state at half-filling[3], where charge localization gives rise to local magnetic moments. The local magnetic moments are thus effectively local particle–hole pairs, and they interact antiferromagnetically with their neighbors and order below the Néel temperature $T_N$ (Fig. 1a). Correspondingly, fluctuations that excite localized charges into the electron–hole continuum above the Mott gap would lead to spatial fluctuations in the size of the magnetic moments, and vice versa (Fig. 1b). It is, however, difficult to detect and exploit this interplay between spin and charge fluctuations because the charge degree of freedom is often frozen in practical Mott materials, like the parent compounds of high-$T_c$ cuprates[4], which are often deep inside the Mott regime. Moreover, the local moments are shielded from the external magnetic field by the antiferromagnetic (AFM) interaction.

$5d$ transition metal oxides provide an intriguing alternative for exploiting such spin-charge interplay[5–8]. In particular, tetravalent iridates can often be considered as effective half-filled single-band systems[9–12] similar to the $3d$ cuprates[13]. As a result, some of their AFM insulating ground state properties can be described by a Hubbard Hamiltonian at the strong coupling limit, i.e., a Mott insulator. On the other hand, the significantly reduced Coulomb interaction and larger extension of the $5d$ orbitals shift these materials toward the Slater regime corresponding to the weak-coupling limit. Both of these two perturbative approaches predict antiferromagnetic insulating ground state, but neither of them provides a complete description of the experimentally observed behaviors[14]. For instance, the charge gap is often found to be reduced from the Mott limit and of a similar size to the magnon bandwidth[15]. Meanwhile, unlike the Slater limit, the insulating behavior and the magnetic moments persist above $T_N$[16–18]. These characters indicate that these materials belong to the crossover regime between the Mott and Slater limits, where neither the Coulomb potential nor the kinetic energy dominates, allowing charge fluctuations to significantly reduce the longitudinal spin stiffness. The Slater–Mott crossover regime is in fact the particle–hole counterpart of the famous Bardeen–Cooper–Schrieffer (BCS)-to-Bose–Einstein condensation (BEC) crossover observed in ultracold-superfluids[19–21]. One may thus anticipate strong spin-

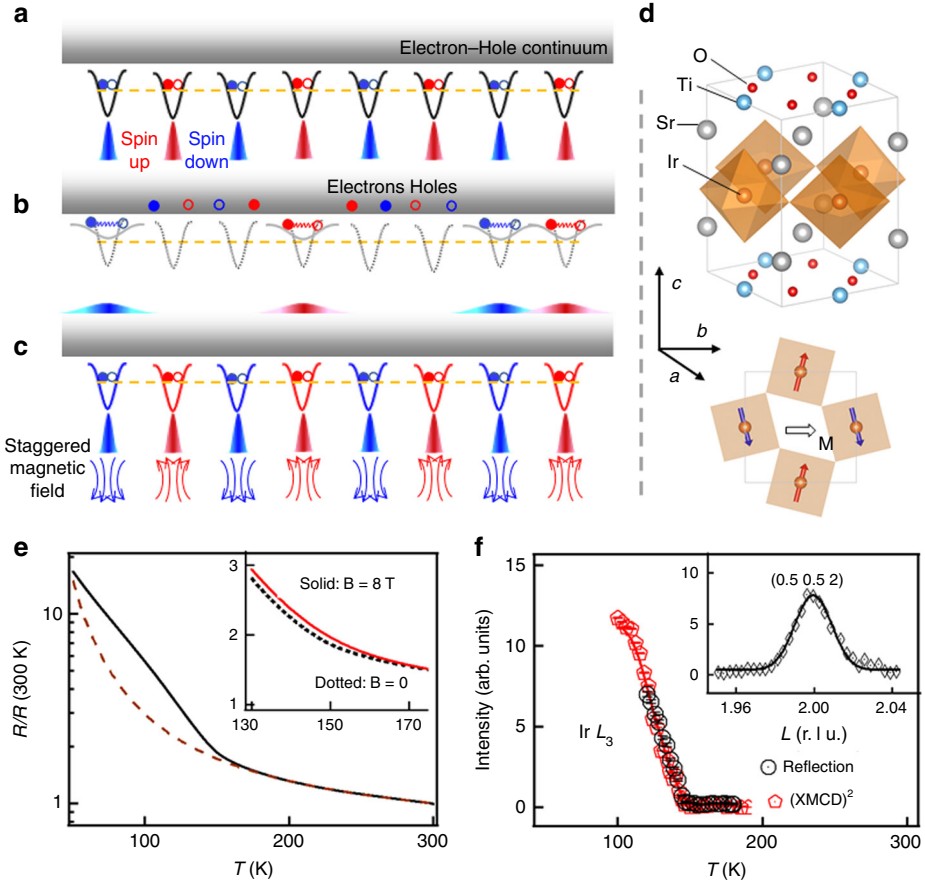

**Fig. 1** Superlattice as a Mott-type AFM insulator. **a–c** Cartoons of a half-filled Hubbard system. **a** Coulomb potential (upward curve) confines one electron–hole pair on each lattice site in an AFM insulating ground state. **b** Magnetic moments decrease with expanded electron-hole pairs or disappear with excitations into the electron-hole continuum. **c** A staggered magnetic field reinforces the staggered moments and the electron-hole pairing. **d** Schematics of the crystal and magnetic structures of the SL. The canted moment **M** is represented as a black arrow. **e** $T$-dependence of the normalized in-plane resistance (solid). It can be well described by the thermal activation model (dash) above $T_N$ (200–300 K), which is extrapolated below $T_N$. Inset shows measurements with and without an in-plane 8 T magnetic field. **f** $T$-dependence of the (0.5 0.5 2) magnetic peak intensity and the XMCD. Inset shows a representative $L$-scan at 10 K. The XMCD is squared since the magnetic peak intensity is proportional to the AFM OP squared.

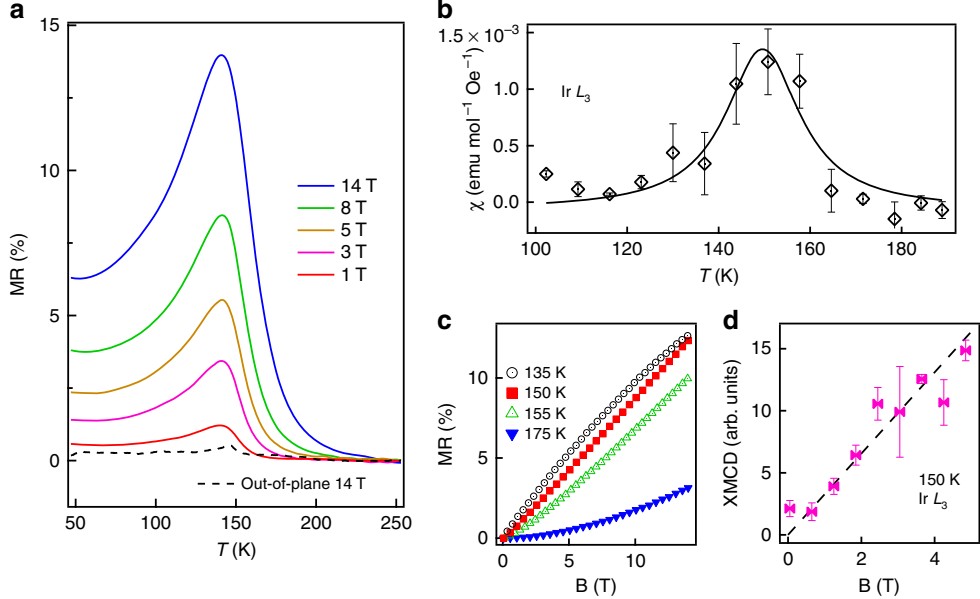

**Fig. 2** MR and XMCD measurements. (**a**) $T$-dependent MR under various in-plane (solid line) magnetic fields and a 14 T out-of-plane (dashed line) magnetic field. (**b**) In-plane uniform susceptibility $\chi$ extracted from the in-plane field-induced XMCD difference (Supplementary Fig. 3). The solid line is a guide to the eye. In-plane magnetic field dependences of MR (**c**) at various temperatures and XMCD (**d**) at 150 K. The error bars come from the statistical averaging every 0.5 T.

charge fluctuations above $T_N$ that are absent in conventional Mott materials and must be considered by including both spin and charge degrees of freedom in the model Hamiltonian.

In this article, we report an experiment-theory-combined investigation on the spin and charge interlay in a square-lattice iridate built as an artificial superlattice (SL) of SrIrO₃ and SrTiO₃, which is well described by a two-dimensional (2D) single-band Hubbard model in the crossover regime between the Mott[3] and the Slater[21] limits. Our results show that, while both limits are adiabatically connected, the strong spin-charge fluctuations in the paramagnetic semiconducting state above $T_N$ hold the key that characterizes the crossover regime. We find that these spin-charge fluctuations can be controlled with an external magnetic field, which couples linearly to the staggered magnetization (Fig. 1c) due to the strong spin–orbit interaction (SOI), and induces a large positive magnetoresistance (MR) above $T_N$. The effects are well reproduced by our calculation, which captures the spatial AFM fluctuations of the paramagnetic state.

## Results

**SrIrO₃/SrTiO₃ superlattice as a Mott-type AFM insulator.** The SL consists of monolayers of SrIrO₃ and SrTiO₃ perovskite stacked alternately on a SrTiO₃ substrate (Fig. 1d). The SL structure is effectively an artificial crystal of Sr₂IrTiO₆ with a confined square lattice of corner-sharing IrO₆ octahedra in the unit cell[22,23]. Moreover, the large SOI of the Ir⁴⁺ ion removes the $t_{2g}$ orbital degeneracy and stabilizes a half-filled $J_{\rm eff} = 1/2$ state (Supplementary Fig. 1), affording a prototypical single-band system[7–10,14,24], which is partially similar to the parent phase of cuprates[13], but with a spin-dependent hopping. The SL exhibits an AFM insulating ground state, including a monotonic exponential resistance increase with reducing temperature $T$ (Fig. 1e) and multiple (0.5 0.5 integer) magnetic reflections (inset of Fig. 1f). The insulating character of the SL is consistent with the dimensional crossover from bulk perovskite SrIrO₃ toward the ultrathin limit[25,26]. The $T$-dependence of the (0.5 0.5 2) peak intensity indicates $T_N \sim 150$ K (Fig. 1f), in agreement with previous reports[22,23]. Furthermore, the AFM order is accompanied

with a weak uniform spontaneous magnetization within the $ab$-plane arising from spin canting due to SOI (Supplementary Fig. 2). The zero-field canting angle is temperature independent and it is determined by the magnitude of the SOI[10,27].

**Anomalous positive MR at $T > T_N$.** Despite the characteristic AFM Mott insulating ground state, the charge transport reveals an anomalous $T$-dependence that cannot be explained within the Mott-Heisenberg scheme. In particular, the insulating behavior is clearly enhanced upon cooling below $T_N$ in comparison to the data above $T_N$ that exhibits a thermally activated behavior with a constant activation energy (Fig. 1e). More interestingly, the resistance can be significantly enhanced near $T_N$ under an in-plane magnetic field (inset of Fig. 1e). This positive MR is in stark contrast to conventional AFM semiconductors and other Mott insulators[28,29], where a negative MR is usually observed due to the field-induced suppression of transverse spin fluctuations. Figure 2a shows the $T$-dependent MR defined as $[R(\mathbf{B}) - R(\mathbf{B} = 0 \, {\rm T})]/R(\mathbf{B} = 0 \, {\rm T})$ under different field strengths. The MR is always positive and displays a strong anomalous behavior where the MR above $T_N$ rapidly increases upon cooling and reaches a maximum around $T_N$, indicative of a large field-induced enhancement of the paramagnetic insulating state. The magnitude of the positive anomalous MR is indeed remarkably large, reaching 14% at 14 T or equivalently ~1%/T, considering the absence of spontaneous long-range magnetic order above $T_N$. In other materials, MR of this magnitude in the paramagnetic state is usually negative and relates to insulator-to-metal phase transition[30,31], highlighting the unusual combination of robust insulating/semiconducting behavior and large positive MR that is present in the SL.

To reveal the role of the external field, we measured x-ray magnetic circular dichroism (XMCD) at the Ir $L_3$-edge. XMCD measures the uniform magnetization, which at zero magnetic field characterizes the canted component of the spontaneous AFM order parameter (OP) as can be seen from its similar $T$-dependence to the AFM Bragg peak (Fig. 1f). The field-induced XMCD variation is thus proportional to the uniform susceptibility $\chi$, which indeed displays a clear maximum around $T_N$

(Fig. 2b). The XMCD shows a positive linear increase as the field scans from 0 to 5 T near $T_N$ (Fig. 2d). Therefore, based on the extracted $\chi$ from XMCD and the thermally activated resistivity above $T_N$, we estimate the MR of ~1%/T corresponds to ~0.12 meV enhancement of the activation energy for every ~$0.02 \times 10^{-3}$ meV increase of the Zeeman energy $\mu_0 \cdot \chi \cdot \mathbf{H}^2$, i.e., a response coefficient of ~6000 in energy scale. In other words, the effect of the external magnetic field is amplified by more than three orders of magnitude in the electronic response due to the strong interplay between spin and charge. Interestingly, since the canting angle $\phi \sim 10^{\circ}$ is determined by a combination of the lattice distortion and the strong SOI[10,27], and is practically unchanged for these field values[32], the measured uniform susceptibility $\chi$ becomes proportional to the staggered susceptibility $\chi^{st}$ near $T_N$ with a proportionality factor $\sin^2\phi \sim 0.03$ (refs. [27,33]). The similar $T$-dependence of $\chi$ and MR suggests that the external field triggers the anomalous charge response near $T_N$ via the large staggered susceptibility. In other words, the MR above $T_N$ is the charge response to the large relative increase of the staggered magnetization induced by the external field. To verify this mechanism, we oriented the field along the $c$-axis where the spin canting is much smaller (Supplementary Fig. 2) and the uniform susceptibility is not sensitive to the staggered susceptibility. The MR becomes strongly suppressed (Fig. 2a), resembling the situation of applying the field to a collinear antiferromagnet. This can indeed be seen from the absence of the MR effect in a square-lattice iridate with a collinear magnetic structure[34].

**Modeling the Slater–Mott crossover regime.** The large anomalous MR in the paramagnetic phase is clearly incompatible with a Mott-Heisenberg regime where charge degrees of freedom are basically frozen because of a charge gap that is much larger than the hopping amplitude. In the opposite weak-coupling limit or Slater regime[21], the charge gap arises from the band reconstruction induced by the AFM ordering and it is directly proportional to the staggered magnetization $\mathbf{M}^{st}$. Correspondingly, an external modulation of $\mathbf{M}^{st}$ is expected to modulate the charge gap causing a charge response that is maximized at $T_N$. The shortcoming of this picture is that the system becomes metallic above $T_N$ and a field-induced gap much smaller than $T_N$ does not necessarily affects the resistivity, which clearly would not account for our observations. The coexistent characteristics of the Slater and the Mott regimes indicate that the observed behavior can only be consistent with the crossover regime (see below). However, modeling the spin-charge interplay and the thermodynamic properties in this intermediate-coupling regime is particularly challenging, especially above $T_N$, because of the lack of a small control parameter[19,20,35]. In other words, to properly capture all the characteristics and the magnetoelectronic response in this regime, one must account for the spatial longitudinal and angular spin fluctuations of the magnetic moments that emerge when the temperature becomes lower than the charge gap, but still well above $T_N$.

We capture these fluctuations by a semi-classical approach where the interaction term of a Hubbard-like model is decoupled via a Hubbard-Stratonovich (HS) transformation (Methods). The motion of thermally activated electrons above the charge gap is then described by an effective quadratic Hamiltonian: fermions propagate under the effect of a fluctuating potential caused by an effective exchange coupling to the underlying HS vector field. The semi-classical approximation arises from the fact that the HS vector field is not allowed to fluctuate along the imaginary time direction[36–38]. Specifically, the effective single-band Hubbard model for the pseudospin-half square-lattice iridates has been

well established and can be written as[27,39,40]

$$H = -t \sum_{\langle ij \rangle} \sum_{\alpha\beta} \left[ c_{i\alpha}^{\dagger} \left( e^{i\varphi \exp(i\mathbf{Q}\cdot\mathbf{r}_i)\sigma^z} \right)_{\alpha\beta} c_{j\beta} + \text{h.c.} \right] + U \sum_j n_{j\uparrow} n_{j\downarrow} - \mathbf{h} \cdot \sum_j \mathbf{s}_j, \tag{1}$$

with the nearest-neighbor $\langle ij \rangle$ hopping amplitude $t$, the onsite Coulomb potential $U$, and the magnetic field $\mathbf{h}$ that couples with the electron spin $\mathbf{s}_j = \frac{1}{2} \sum_{\alpha\beta} c_{j\alpha}^{\dagger} \boldsymbol{\sigma}_{\alpha\beta} c_{j\beta}$. The wave vector $\mathbf{Q} = (\pi, \pi)$ distinguishes the two sublattices and the phase factor $e^{i\varphi \exp(i\mathbf{Q}\cdot\mathbf{r}_i)\sigma^z}$ represents the spin-dependent hopping enabled by SOI and octahedral rotation (Fig. 3)[41]. Unlike the usual spin-half Hubbard model, this phase factor renders complex hopping integrals for different spins due to the spin–orbit-entangled $J_{eff} = 1/2$ wavefunctions. It is important to note that the in-plane spin canting in the AFM ground state is ultimately driven by this spin-dependent hopping and the angle $\varphi$ of the phase factor determines the canting angle $\phi$ at zero field[42,43]. At finite fields, it determines the ratio of the uniform susceptibility and the staggered susceptibility. Therefore, the spin-dependent hopping allows the external magnetic field to couple linearly with the staggered magnetization through the uniform component at any temperature. To reveal this point, we perform a staggered reference frame transformation, $\tilde{c}_{j\alpha} = \sum_{\beta} \left[ e^{-i\frac{\varphi}{2}\exp(i\mathbf{Q}\cdot\mathbf{r}_j)\sigma^z} \right]_{\alpha\beta} c_{j\beta}$, to convert the global spin frame into the local spin frame depicted in Fig. 3. In the new reference frame, the Hamiltonian becomes:

$$H = -t \sum_{\langle ij \rangle, \sigma} \left[ \tilde{c}_{i\sigma}^{\dagger} \tilde{c}_{j\sigma} + \text{h.c.} \right] + U \sum_j \tilde{n}_{j\uparrow} \tilde{n}_{j\downarrow} - h^z \sum_j \tilde{s}_j^z - \cos\varphi \mathbf{h}^{\perp} \cdot \sum_j \tilde{\mathbf{s}}_j^{\perp} + \sin\varphi (i\sigma^y \mathbf{h}^{\perp}) \cdot \sum_j \tilde{\mathbf{s}}_j^{\perp} e^{i\mathbf{Q}\cdot\mathbf{r}_j}, \tag{2}$$

where $\tilde{s}_j^z$ and $\tilde{\mathbf{s}}_j^{\perp}$ are the out-of-plane and in-plane components of the transformed spin $\tilde{\mathbf{s}}_j = \frac{1}{2} \sum_{\alpha\beta} \tilde{c}_{j\alpha}^{\dagger} \boldsymbol{\sigma}_{\alpha\beta} \tilde{c}_{j\beta}$, and $h^z$ and $\mathbf{h}^{\perp}$ are the out-of-plane and in-plane components of the external field. Interestingly, the phase factor is gauged away by this transformation of the reference frame[10,27,32,44,45], which uncovers the Hubbard model with the usual spin-independent hopping and a

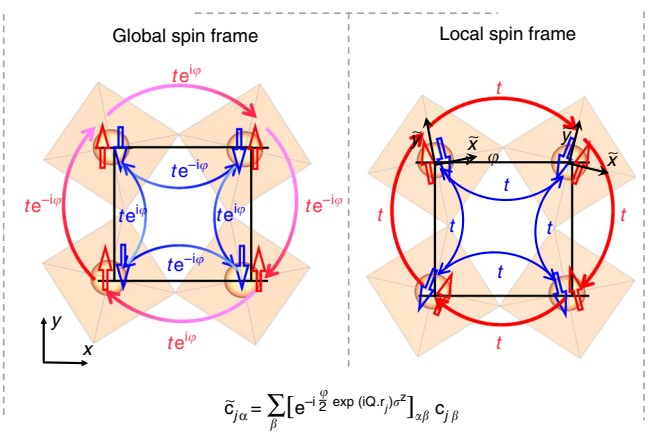

$$\tilde{c}_{j\alpha} = \sum_{\beta} [e^{-i\frac{\varphi}{2}\exp(i\mathbf{Q}\cdot\mathbf{r}_j)\sigma^z}]_{\alpha\beta} c_{j\beta}$$

**Fig. 3** Charge hopping in spin-up (red) and spin-down (blue) in different spin frames. In the global spin frame (left panel), charge hopping bears an alternating phase factor when circling around the square lattice. This phase factor is gauged away in the rotated local spin frame (right panel), leading to an isotropic Hubbard model[27,33]. The annihilation operators $\tilde{c}_{j\alpha}$ in the local frame are transformed from $c_{j,\beta}$ in the global frame according to the shown transformation.

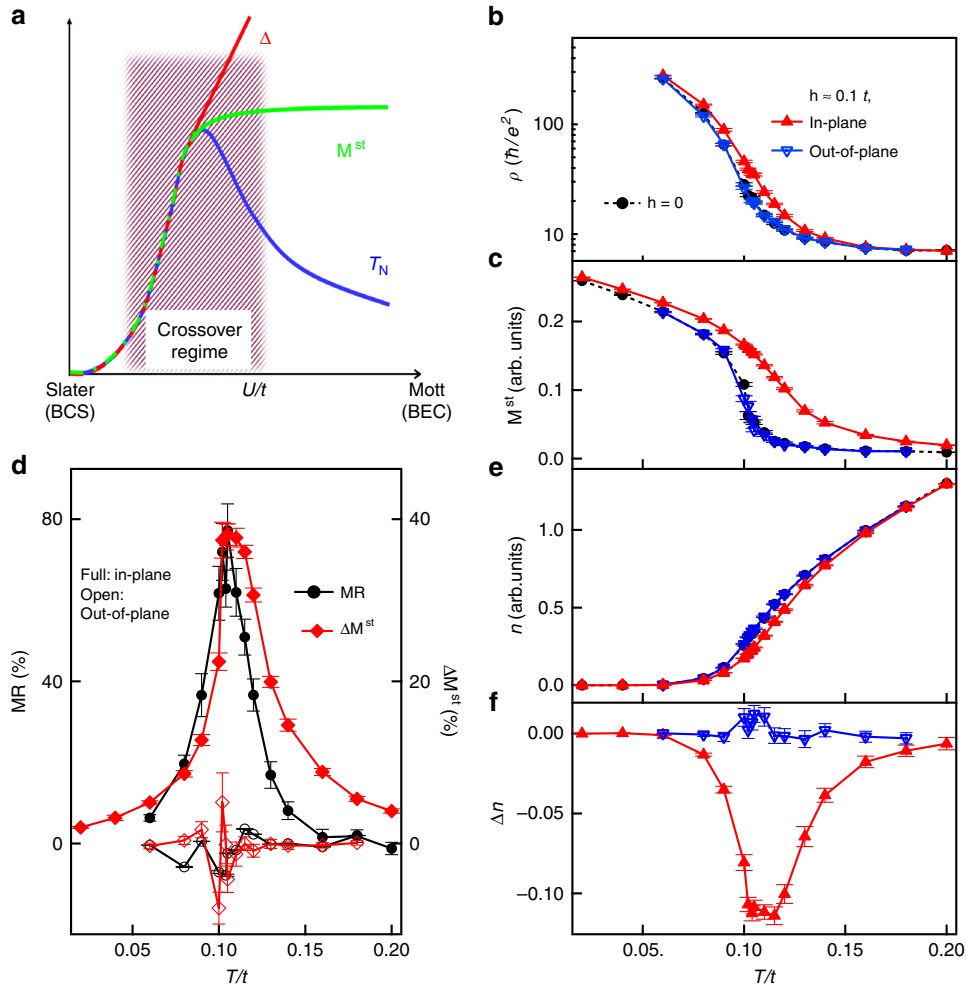

**Fig. 4** Theoretical calculations. **a** A generic phase diagram of the Slater–Mott crossover. The charge gap $\Delta$ is shown with $T_N$ and $\mathbf{M}^{st}$ as a function of $U/t$. $T$-dependent resistivity (**b**), $\mathbf{M}^{st}$ (**c**) and carrier density $n$ (**e**) calculated at zero field (black circles), in-plane field (red up triangles) and out-of-plane field (blue down triangles) for $\mathbf{h} \approx 0.1t$ and $U = 3t$. $T$-dependent MR (circles) and field-induced $\mathbf{M}^{st}$-variation (diamonds) (**d**), and field-induced $n$-variation (**f**) under an in-plane (full) and out-of-plane (open) field. The error bars represent the statistical error.

linear coupling between the external field and the staggered magnetization scaled by $\sin\varphi$.

Figure 4b shows the longitudinal resistivity $\rho$ computed with the Kubo formula[46] on a square lattice of $64 \times 64$ atoms for the intermediate-coupling strength and Hamiltonian parameters relevant to the present SL (Methods). $\rho$ indeed displays an insulating exponential increase when decreasing temperature. From the $T$-dependent $\mathbf{M}^{st}$, we identified a non-zero AFM transition temperature around $T/t \sim 0.1$ (Fig. 4c) after including the easy-plane anisotropy that arises from the Hund's coupling (Methods). Under an in-plane magnetic field, $\mathbf{M}^{st}$ is clearly enhanced and it becomes finite above $T_N$. The response (Fig. 4d) exhibits a $T$-dependence typical of the staggered susceptibility, similar to the XMCD data. The calculated MR at temperatures above $T_N$ is also positive and it reaches a maximum at the AFM transition in agreement with the experimental observation (Fig. 2a). In stark contrast, $\mathbf{M}^{st}$ and $\rho$ remain almost unchanged under the effect of an out-of-plane field due to the lack of the staggered field effect that is expected from Eq. (2). We note that we have used a much larger field than the experimental value because of limitations in the numerical accuracy of our unbiased stochastic estimator of the conductivity. Additionally, extrinsic effects that may dominate the conductivity well below $T_N$, such as magnetic domains or domain walls[47,48], cannot be captured by

our model. Nevertheless, the remarkable agreement between experiment and theory at temperatures above and near $T_N$, where magnetic domains are absent, demonstrates that the MR is an indirect electronic probe of the staggered susceptibility whenever the field couples linearly to $\mathbf{M}^{st}$.

To gain microscopic insights, we have also computed the $T$-dependence of the thermally activated carriers $n$ (free particles/holes) using model (1). As shown in Fig. 4e, the suppression of $n$ accelerates upon cooling toward the AFM transition, below which $n$ quickly drops to zero. This behavior illustrates the unique character of the Slater–Mott crossover regime where a large number of particle–hole pairs are pre-formed well above $T_N$ with a fluctuating coherence length of several lattice spaces. This size fluctuation and the corresponding longitudinal spin fluctuations are critically suppressed upon cooling toward $T_N$. By inducing a finite $\mathbf{M}^{st}$, the in-plane magnetic field further suppresses $n$ through increasing the binding energy of the particle–hole pairs. This tunability maximizes near $T_N$ (largest AFM susceptibility), while it decreases upon rotating the field to the out-of-plane direction (Fig. 4f). This analysis uncovers the role of the SOI, which enables a linear coupling between the uniform field and the staggered magnetization and therefore a relatively large positive anomalous MR due to the large longitudinal staggered susceptibility of the Slater–Mott crossover regime.

## Discussion

From the experimental results and the theoretical simulations, we can conclude that the two basic ingredients of the positive anomalous MR are (1) the strong interplay between spin and charge in the paramagnetic state of the Slater–Mott crossover regime, and (2) the spin-dependent hopping enabled by the strong SOI and the lattice structure, emerging as ferromagnetic canting in the ground state. While the former ingredient is present in many correlated systems, the latter one is subject to multiple competing interactions, such as easy-plane vs. easy-axis anisotropy. The structure of our SL is designed to minimize such competition as the octahedral network is rotated in the same way among all $IrO_6$ layers (Fig. 1d)[49]. For comparison, the pseudospin-half iridate $Sr_2IrO_4$ has a much more complicated layered structure and a magnetic unit cell that contains four $IrO_6$ layers with a substantial and nontrivial interlayer interaction that favors cancellation of the canted moments[33]. These differences explain why the MR of $Sr_2IrO_4$ is negative and governed by the transverse spin fluctuations[50–53].

Note that, although the magnitude of the observed anomalous MR is smaller than the GMR[54] and CMR effects[31] of magnetic metals, distinct MR effects often indicate a new physical mechanism, like the one present in the recently discovered spin-Hall MR effect[55,56]. In our case, the sensitivity of MR to the longitudinal spin fluctuations provides an efficient electronic probe of the usually elusive staggered susceptibility of Mott-type insulating materials. This magnetoelectronic effect provides a mechanism that is fundamentally distinct from that in itinerant magnets and conventional magnetic semiconductors[29,57], where the magnetic moments and the carriers are two separate subsystems and orientation control of the OP is the dominant mechanism for modifying the carrier transport[28]. In contrast, the spin and the charge are necessarily provided by the same electrons in Hubbard-like systems. The Mott semiconductor that emerges in the Slater–Mott crossover regime has the best performance near $T_N$, which is expected to be maximized in this regime (Fig. 4a). Moreover, since longitudinal fluctuations have higher frequency than the transverse fluctuations[58], the Slater–Mott crossover regime may enable high-speed electronics. If combined with orientation control of the AFM moments[59,60], it may pave a way to merge the information processing and storage functionalities in a single material with enhanced device density.

In summary, we have demonstrated the ability to control resistivity by exploiting the strong interplay between the staggered magnetization and the effective Coulomb potential in a quasi-two-dimensional AFM Mott insulator. The strong longitudinal spin fluctuations in the Slater–Mott crossover regime are exposed to external field by SOI and enable a significant MR that peaks around $T_N$. This magnetoelectronic effect has not been observed in strongly correlated Mott insulators, such as cuprates[61,62], or in weakly correlated Slater insulators[63], highlighting the nontrivial spin-charge fluctuations of the crossover regime and the importance of strong SOI. The work thus opens a door for designing AFM electronics in spin–orbit-entangled correlated materials.

## Methods

**Sample synthesis**. The superlattice of $[(SrIrO_3)_1/(SrTiO_3)_1]$ was fabricated by means of pulsed laser deposition on a single crystal $SrTiO_3$ (001) substrate. The deposition process was in-situ monitored through an equipped reflection high-energy electron diffraction unit. This guarantees an accurate control of the atomic stacking sequence (60 repeats). Optimized growth temperature, oxygen pressure, and laser fluence are 700 °C, 0.1 mbar and 1.8 J/cm², respectively. Detailed structural characterizations can be found in ref. [23].

**Materials characterizations**. The sample magnetization was characterized with a Vibrating Sample Magnetometer (Quantum design). In-plane and out-of-plane remnant magnetization were recorded during zero-field warming process. The sample resistance was measured by using the standard four-probe method on a physical properties measurement system (PPMS, Quantum design) and a PPMS Dynacool. For the in-plane MR measurements, the magnetic field is applied along the STO (100) direction. The X-ray absorption (XAS) and X-ray magnetic circular dichroism (XMCD) data was collected around the Ir $L_3$- and $L_2$-edges on beamline 4IDD at the Argonne National Laboratory, which features a high magnetic field strength of 6 T. For these measurements, the samples were monitored in a grazing incidence geometry and a fluorescence yield mode was adopted. Magnetic scattering experiments near Ir $L_3$-edge were performed on beamline 6IDB, at the Advanced Photon Source of Argonne National Laboratory. A pseudo-tetragonal unit cell $a \times a \times c$ ($a = 3.905$ Å, $c = 3.954$ Å)[23] was used to define the reciprocal lattice notation.

**Numerical simulation**. To account for the easy-plane anisotropy arising from Hund's coupling, the following term is included in the total Hamiltonian[10]:

$$H_A = -\Gamma_1 \sum_{\langle ij \rangle} \tilde{s}_i^z \tilde{s}_j^z \pm \Gamma_2 \sum_{\langle ij \rangle} \left( \tilde{s}_i^x \tilde{s}_j^x - \tilde{s}_i^y \tilde{s}_j^y \right), \qquad (3)$$

where the $+$ $(-)$ sign is taken for bonds along the $x$ ($y$) direction.

To study the electrical response, we first perform a Hubbard–Stratonovich transformation to the Hamiltonian $H + H_A$, which gives the spin fermion Hamiltonian[36–38]:

$$
\begin{aligned}
H_{SDW} = & -t \sum_{\langle ij \rangle} \sum_{\alpha\beta} \left[ c_{i\alpha}^\dagger \left( e^{i\varphi \exp(i\mathbf{Q}\cdot\mathbf{r}_i)\sigma^z} \right)_{\alpha\beta} c_{j\beta} + \text{h.c.} \right] \\
& - 2U \sum_j \mathbf{m}_j \cdot \mathbf{s}_j + U \sum_j \left| \mathbf{m}_j \right|^2 - \mathbf{h} \cdot \sum_j \mathbf{s}_j \\
& - \Gamma_1 \sum_{\langle ij \rangle} \left( m_i^z s_j^z + m_j^z s_i^z - m_i^z m_j^z \right) \\
& \pm \Gamma_2 \sum_{\langle ij \rangle} \left( m_i^x s_j^x + m_j^x s_i^x - m_i^y s_j^y - m_j^y s_i^y - m_i^x m_j^x + m_i^y m_j^y \right),
\end{aligned}
\qquad (4)
$$

where the local auxiliary field $\mathbf{m}_i$ is a classical vector in $\mathbb{R}^3$.

The equilibrium configurations of $\mathbf{m}_i$ are sampled via the stochastic Ginzburg-Landau (GL) relaxation dynamics[38,64,65]. In $5d$ iridates, the strong SOI plays a unique role in competing with the electron correlation, giving rise to the strong locking of crystal lattice and magnetic moments[10,66]. In other words, the Ir magnetic moment strictly follows the rotation of $IrO_6$ octahedral rotation[10]. By assuming an Ir–O bond length similar to that in the $Sr_2IrO_4$ single crystal[11], the pseudo-tetragonal unit cell of the SL gives an octahedral rotation angle ~ 10°. A value $\varphi = 10°$ was therefore adopted for the numerical simulation. In the simulation, we use a $64 \times 64$ square lattice with $t = \frac{1}{\cos\varphi} \approx 1.02$, $U = 3$, $\Gamma_1 = \Gamma_2 = 0.05$, and three choices of magnetic field $\mathbf{h} = \{(0, 0, 0), (0.1, 0, 0), (0, 0, 0.1)\}$. The GL dynamics with damping parameter $\alpha = 0.1$ is integrated using the Heun-projected scheme[67] with time-step $\Delta\tau = 0.01$. The molecular torques on $\mathbf{m}_i$ are obtained by integrating out the electrons at each time step using the kernel polynomial method and gradient based probing[38,68–72], with $\mathbf{M} = 500$ Chebyshev moments and $\mathbf{R} = 128$ random vectors.

After obtaining the equilibrium spin configurations, we use Kubo formula[46] to evaluate the longitudinal conductivity, by diagonalizing Equation (4) exactly. A Lorentzian broadening factor $\eta = 1/64$ is used in the Kubo formula calculation. For each temperature, we average the longitudinal conductivity over 20 snapshots, separated by at least twice of the auto correlation time (For example, for $\mathbf{h} = (0, 0, 0)$, $T = 0.102$, the separation between two snapshots is $3 \times 10^4$ integration time steps).

The $\mathbf{M}^{st}$ in the main text is defined as

$$(\mathbf{M}^{st})^2 \equiv \frac{1}{N} \langle \bar{\mathbf{s}}_\mathbf{Q} \cdot \bar{\mathbf{s}}_{-\mathbf{Q}} \rangle, \qquad (5)$$

where $N = 64 \times 64$ is the total number of lattice sites, and $\bar{\mathbf{s}}_\mathbf{Q}$ is the Fourier transform of $\bar{\mathbf{s}}_j$ defined in the main text.

## Data availability

The authors declare that the data supporting the findings in the current study are available from the corresponding author upon reasonable request.

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

## Acknowledgements

The authors acknowledge experimental assistance and discussion with H. D. Zhou, H. X. Xu, H. Suwa, Eun Sang Choi, C. Rouleau, Z. Gai, J. K. Keum, and N. Traynor. J.L. acknowledges the support by the start-up fund and the Science Alliance Joint Directed Research & Development Program at the University of Tennessee, the National Science Foundation under Grant No. DMR-1848269, and the DOD-DARPA under Grant No. HR0011-16-1-0005. J.L. and C.D.B. acknowledge the support by the Organized Research Unit Program at the University of Tennessee. Z.W. and C.D.B. are supported by funding from the Lincoln Chair of Excellence in Physics. M.P.M.D. and D.M. are supported by the U.S. Department of Energy, Office of Basic Energy Sciences, Early Career Award Program under Award Number 1047478. J.S. and J.-H.C. are supported by the Air Force Office of Scientific Research Young Investigator Program under Grant FA9550-17-1-0217. K.B. is supported by the LDRD program at Los Alamos National Laboratory. A portion of the work was conducted at the Center for Nanophase Materials Sciences, which is a DOE Office of Science User Facility. Use of the Advanced Photon Source, an Office of Science User Facility operated for the U.S. DOE, OS by Argonne National Laboratory, was supported by the U.S. DOE under Contract No. DE-AC02-06CH11357. This work used resources of the Compute and Data Environment for Science (CADES) at the Oak Ridge National Laboratory, which is supported by the Office of Science of the U.S. Department of Energy under Contract No. DE-AC05-00OR22725. This research used resources of the Oak Ridge Leadership Computing Facility, which is a DOE Office of Science User Facility supported under Contract DE-AC05-00OR22725.

## Author contributions

J.L. conceived and directed the study. L.H., J.Y. and D.M. undertook sample growth and characterization. L.H., J.Y., D.M., J.W.K. and P.J.R. performed magnetic scattering measurements. L.H., J.Y., G.F., Y.S.C. and D.H. conducted XMCD measurements. L.H., J.Y., J.S. and J.H.C performed transport measurements. L.H. and J.L. analyzed data. Z.W., K.B. and C.D.B. performed the numerical simulations. L.H., Z.W., M.P.M.D., C.D.B. and J.L. wrote the manuscript.

## Competing interests

The authors declare no competing interests.
