## [Peer Review File · Nature Communications]

Reviewers' Comments:

Reviewer #1:

Remarks to the Author:

This paper discusses the magnetoresistance of SrIrO₃/SrTiO₃ superlattices driven by the coupling of the magnetic field to the canted moment in an antiferromagnetic insulator. The authors present a range of experimental data (transport and XMCD) and model 2D Monte Carlo calculations in support of this idea. Given the very topical interest in spin-orbit coupled oxides, oxide heterostructures, and the joint expt-theory results reported here, I am somewhat inclined to accept this paper for NComms. However, it would be useful to address the following two issues, due to which I have some reservations.

1. A uniform AFM insulator would be expected to have high resistivity. However in a real system, we might expect magnetic domains where the canted moment can point along x or y, with domain walls being more conducting. A magnetic field could result in domain selection, which would suppress such domain walls and enhance resistivity. It would be useful to add a few sentences clarifying the role (or absence of role) of such domains in the observed magnetotransport.

2. Why heterostructures? Shouldn't Sr₂IrO₄ (which are also Slater-Mott insulators with intermediate correlations) show the same physics? Has this been observed already in Sr₂IrO₄ crystals or how is this study really distinct? Is it just one more system?

Reviewer #2:

Remarks to the Author:

This paper presents a detailed study of the magneto-transport of SrIrO₃/SrTiO₃ superlattice. Combined with the magnetoresistivity measurement, X-ray spectroscopy and model calculation, the authors associate the positive magnetoresistivity with the antiferromagnetic spin fluctuation inherent to the Slater-Mott crossover. Although the paper presents some interesting concepts, I could not find any outstanding features in the experimental result. The main result (magnetoresistivity and XMCD) appears to be conventional and to require the no special concept. In addition, there are many open questions about the data analysis and interpretation to support the author's conclusion. I therefore do not recommend publication of this manuscript in Nature Communications.

1. The positive magnetoresistivity is observed nearby the magnetic transition temperature (T_N), but also substantially remains well below T_N (Fig. 2a). On the other hand, the model calculation (Fig. 4d) shows that the magnetoresistivity is observed only nearby T_N . The theory appears to fail to explain the experimental result even in qualitative level. Authors should append further explanation.

2. The reason why the XMCD can detect the antiferromagnetic order parameter (staggered susceptibility) is not clear. In general, XMCD is scaled to the uniform (ferromagnetic) susceptibility. In this sense, the resonant magnetic scattering at (0.5, 0.5, 2) would be related to the antiferromagnetic order parameter. It is not clear why authors associate the magnetoresistivity with the XMCD rather than the resonant magnetic reflection to discuss the antiferromagnetic order parameter.

3. Authors assume that the canting angle ϕ is unchanged even nearby antiferromagnetic transition temperature. The reason is not clear. The uniaxial anisotropy is usually well defined when the magnetic order parameter is sufficiently large (longitudinal magnetic fluctuation is negligible). Moreover, regarding this point, it is not comprehensible that the magnetic field induces a finite antiferromagnetic order parameter above T_N .

4. In the discussion part, authors call this magnetoresistivity as "magnetoelectric effect". However, conventionally, the magnetoresistivity is not termed as magnetoelectric effect.

Reviewer #3:

Remarks to the Author:

The authors report a magnetoresistance (MR) measurement in the antiferromagnetic Mott insulator superlattice SrIrO₃/SrTiO₃. Distinct from the negative MR found in other Mott insulators, the MR in the SrIrO₃/SrTiO₃ superlattice is positive which shows a maximum at the Neel temperature. The positive MR observed is interpreted with the magnetic field suppression of the spin fluctuation and a theoretical calculation is included in the manuscript. I hope the authors can address the following issues, which will help me to judge whether the present work deserves the publication in Nature Communications.

1. The magnitude of the MR in reported in this work is about 1%/T which is strongest near the Neel temperature. Have to say that such a MR magnitude is not really "large" compared to the MR in the TMR and GMR device which are applicable. However, MR with small magnitude can still be indispensable for the characterization of material properties. For example: the recently discovered spin Hall magnetoresistance [H. Nakayama et al., PRL 110, 206601 (2013)][D. Hou et al., PRL 118, 147202 (2017)], can be used to probe the spin diffusion length, spin Hall angle, the spin mixing conductance and even the interface magnetic moment directions in spite of the 0.01% MR magnitude. So, are the authors able to evaluate some parameters in this SrIrO₃/SrTiO₃ superlattice? This is also an important reference for this reviewer to judge the novelty of the present work

2. I wonder whether the authors can make some argument about other systems which probably show a positive MR due to the same mechanism. If a phenomenon can be observed in a variety of materials, it is very likely to be important and of general interest.

3. In line 54, it reads "...to control and practical Mott materials," the "and" should be "of", right?

4. I suggest the authors to specify the orientation of the STO substrates (is it STO(001)?).

5. Does the direction of magnetic field in the a-b plane matter for the MR result? And please specify which direction the field is applied during the measurement.

6. For the argument made in line 124-126, please add more reference to support it, otherwise it is very hard to judge for general audience.

7. I wonder whether it is necessary to emphasize the "magnetic field is equivalent to a staggered field", because this argument is too general and it is essentially the vector addition.

**To Reviewer #1:**

*This paper discusses the magnetoresistance of SrIrO₃/SrTiO₃ superlattices driven by the*
*coupling of the magnetic field to the canted moment in an antiferromagnetic insulator. The*
*authors present a range of experimental data (transport and XMCD) and model 2D Monte Carlo*
*calculations in support of this idea. Given the very topical interest in spin-orbit coupled oxides,*
*oxide heterostructures, and the joint expt-theory results reported here, I am somewhat inclined*
*to accept this paper for NComms. However, it would be useful to address the following two*
*issues, due to which I have some reservations.*

**Response:**

We thank the referee for the careful review of our manuscript. We appreciate her/his
recognition of the value of our work and recommendation for publication in *Nature*
*communications*. Below we address each of the points listed in the report.

*1. A uniform AFM insulator would be expected to have high resistivity. However in a real system,*
*we might expect magnetic domains where the canted moment can point along x or y, with*
*domain walls being more conducting. A magnetic field could result in domain selection, which*
*would suppress such domain walls and enhance resistivity. It would be useful to add a few*
*sentences clarifying the role (or absence of role) of such domains in the observed*
*magnetotransport.*

**Response:**

We appreciate this comment. It is indeed true that domain wall conduction may play a
role in the transport properties below the Néel temperature of some antiferromagnetic insulating
materials, like BiFeO₃ and YMnO₃. Due to the complex mechanism, resolving domain wall
conduction and its contribution to the average macroscopic conduction is highly nontrivial and
requires precise control of the domain walls, as well as microscopic probes, such as a
combination of domain imaging and nanoscale transport measurements. This is certainly beyond
the scope of our manuscript, which focuses on the physics in the regime *above* T_N . We note that
previous studies of related iridate system Sr₂IrO₄, shows some gap inhomogeneity [*Sci. Rep.* **3**,
3073 (2013)], even there is no evidence of conducting domain walls. But such domain structure
and domain conduction are nonetheless interesting directions moving forward.

In order to provide some insights, we performed a rough estimation of the domain wall
contribution by cooling the sample in different magnetic fields and then comparing its resistance
as a function of in-plane magnetic field. The hypothesis of this method is that field-cooling (FC)
and zero-field-cooling (ZFC) procedures lead to different domain populations and different
domain wall densities, which is a reasonable assumption for magnetic materials. Since the zero-
field resistivity within each domain should be the same, the difference between the FC and ZFC
resistances should reflect the domain wall contribution. Figure R1 shows the data at 150 K and
110 K, where we normalize the FC and ZFC resistance by the resistance at 8T. Below T_N (at 110
40 K), a small yet observable difference between the ZFC and FC resistances is found near zero
field and can be attributed to the domain wall contribution, which is only 0.2% and small
compared with the total MR of 8% at 8 T. The difference between ZFC and FC resistances
extends to finite fields as the ZFC resistance shows a wiggling shape, which is absent in the FC
resistance and can be attributed to hysteretic domain switching in the applied field, further

confirming the domain-driven origin of this behavior. Although the larger zero-field normalized
resistance in the ZFC curve suggests that domain walls are less conducting than the domains, the
0.2% difference is actually close to the measurement accuracy limited by the temperature
reproducibility of the PPMS in different thermal cycles. Future microscopic investigation is
needed to pinpoint the domain wall conduction. *Nevertheless, this sets the upper limit of the*
*domain wall contribution to our MR measurement to be the order of 0.1%.* And all the data
represented in the manuscript were obtained after FC. More importantly, this resistance
difference is absent at 150 K, confirming that spontaneous long-range order is absent and that
there is no domain effect contribution to the MR near this temperature and above. We now added
the data and related discussions in the supplemental materials.

**Figure R1.** In-plane magnetic field dependent normalized resistance at different temperatures.

The resistance was measured after cooling the sample under zero magnetic field (ZFC) or an 8 T

in-plane magnetic field (FC). The low-field resistance difference at 100 K is highlighted in a blue
rectangle.

*2. Why heterostructures? Shouldn't Sr₂IrO₄ (which are also Slater-Mott insulators with*
*intermediate correlations) show the same physics? Has this been observed already in Sr₂IrO₄*
*crystals or how is this study really distinct? Is it just one more system?*

**Response:**

We should clarify that the physics of longitudinal fluctuations in the *Slater-Mott*
*crossover* regime is general and many physical systems show signatures of these fluctuations in
the absence of the applied field. However, *the observation of the magnetoelectronic response*
*requires a linear coupling between the external uniform magnetic field and the staggered*
*moment*, which is absent in most compounds. Because our tailored heterostructure enables this
linear coupling in the absence of other competing effects, we can experimentally demonstrate for
the first time how longitudinal fluctuations produce a positive anomalous magneto-resistance.

We have not studied Sr₂IrO₄ bulk crystals and whether or not these materials show
similar effects represents an important question for future studies – so thank you for raising this
interesting point. Based on the results published in the literature, we discuss the differences with
Sr₂IrO₄:

**a)** Sr₂IrO₄ is a pure AFM. Specifically, in zero field, the canted magnetic moment of each
iridate perovskite layer is antiferromagnetically aligned with the canted component on the
adjacent layer, giving rise to a zero net magnetization for Sr₂IrO₄ [*Science* **323**, 1329
(2009)]. The canted magnetization components of different layers align parallel with each

other only when the applied magnetic field is stronger than the critical field of the
metamagnetic transition [*Phys. Rev. B* **57**, 11039 (1998)]. Interestingly, this
metamagnetic transition of Sr₂IrO₄ is accompanied with a huge negative
magnetoresistance (>10%) [*Phys. Rev. B* **84**, 100402 (2011); *Adv. Mater.* **30**, 1805564
(2018)]. While the precise microscopic origin of these behaviors is still unresolved, it is
clear that the interlayer hopping in Sr₂IrO₄ must be significant to generate such electronic
response during the metamagnetic transition [*Phys. Rev. B* **94**, 224420 (2016)].

To account for the complicated magnetic structure of Sr₂IrO₄, one may consider an
effective 2D model, where the interlayer hopping is included as an effective exchange (or
molecular) field that opposes the external field [*Phys. Rev. B* **94**, 224420 (2016)]. At
$T > T_N$, which is the focus of our study, the external field induces an AFM order via the
staggered field enabled by the spin-orbit coupling in all layers with parallel canted
moments. However, since the interlayer hopping favors opposite canted moment
alignment between the layers, its effective exchange field would have to point in the
opposite direction to the external field. These two effects would thus cancel each other.
The situation is probably even more complicated in a 3D model for Sr₂IrO₄ with a
magnetic unit cell that contains four layers.

In contrast, the heterostructure has a net magnetization with a “clean” canted AFM
ground state without any internal competing effect. The finite spontaneous magnetization
is consistent with the fact that the interlayer hopping favors parallel alignment of the spin
canting of all layers. Thus, the layers are identical to each other and a 2D Hubbard model
can be effectively applied.

**b)** If the above mentioned effects were absent, a similar charge response would be
expected in Sr_2IrO_4 , since Sr_2IrO_4 is also believed to be in the *Slater-Mott crossover*
regime. We note, however, that there is no clear resistance anomaly around T_N in Sr_2IrO_4 ,
which means that the longitudinal AFM fluctuations give a small contribution to the
overall DC transport properties. Indeed, a recent in-plane MR study on Sr_2IrO_4 reveals a
dominant role of transverse spin fluctuations [*Adv. Funct. Mater.* 1706589 (2018)], which
leads to a negative MR value. In contrast, as shown in Fig. 1e of the main text, the
heterostructure displays a clear resistance anomaly around T_N , which is the first zero-field
indication of a substantial contribution of the longitudinal AFM fluctuations. This
difference suggests that, although both systems are in the Slater-Mott crossover regime,
Sr_2IrO_4 is much closer to the Mott limit than the heterostructure.

In connection to this point, we note that another layered iridate, $\text{Sr}_3\text{Ir}_2\text{O}_7$, also exhibits a
resistance anomaly around T_N [*Phys. Rev. B* **66**, 214412 (2002)] and it is likely to be in the
crossover regime as well. However, $\text{Sr}_3\text{Ir}_2\text{O}_7$ has a *c*-axis collinear AFM structure with zero net
magnetization [*Phys. Rev. Lett.* **109**, 037204 (2012)], indicating a strong intra-bilayer hopping
that is necessarily spin-dependent and gives rise to the strong *c*-axis anisotropy. Like in the case
of Sr_2IrO_4 , this intra-bilayer effect acts as a strong effective AFM exchange field along the *c*-axis
that makes the system much less susceptible to the uniform external field. To solve this issue, we
built another heterostructure, $[(\text{SrIrO}_3)_2/(\text{SrTiO}_3)_1]$, which is a bilayer system and also has a *c*-
axis AFM structure similar to $\text{Sr}_3\text{Ir}_2\text{O}_7$, but with a spontaneous in-plane net moment due to
canting [*Phys. Rev. Lett.* **114**, 247209 (2015); *Phys. Rev. Lett.* **119**, 027204 (2017); *Sci. Rep.* **9**,
4263 (2019)]. The canting is caused by additional octahedral tilts, which are absent in $\text{Sr}_3\text{Ir}_2\text{O}_7$
and create a spin-dependent intra-bilayer hopping channel similar to that within the *ab*-plane

discussed in the manuscript. As a result, an in-plane external field can induce an effective c -axis
staggered field on top of the internal one. As shown in Fig. R2, the MR under an in-plane field is
positive and has a maximum around T_N , which is consistent with the result for the
$[(\text{SrIrO}_3)_1/(\text{SrTiO}_3)_1]$ superlattice albeit with smaller magnitude.

**Figure R2.** (a) Temperature dependent in-plane MR of $[(\text{SrIrO}_3)_2/(\text{SrTiO}_3)_1]$ grown on a
$\text{SrTiO}_3(001)$ substrate. (b) Temperature dependence of the in-plane zero-field magnetization.

We have added a paragraph *“From the experimental results and the theoretical*
*simulations, we can conclude that the two basic ingredients of the positive anomalous MR are:*
*(1) the strong interplay between longitudinal spin fluctuations and particle-hole pairing strength*
*in the paramagnetic state of the Slater-Mott crossover regime, and (2) the spin-dependent*
*hopping enabled by the strong SOI and the lattice structure, emerging as ferromagnetic canting*
*in the ground state. While the former ingredient is present in many correlated systems, the latter*
*one is subject to multiple competing interactions, such as easy-plane vs. easy-axis anisotropy.*
*The structure of our SL is designed to minimize such competition as the octahedral network is*
*rotated in the same way among all IrO₆ layers (Fig. 1d)⁴². For comparison, the pseudo-spin-half*
*iridate Sr₂IrO₄ has a much more complicated layered structure and a magnetic unit cell that*
*contains four IrO₆ layers with a substantial and nontrivial interlayer interaction that favors*
*cancellation of the canted moments²⁶. These differences explain why the MR of Sr₂IrO₄ is*
*negative and governed by the transverse spin fluctuations⁴³⁻⁴⁶.”* in the discussion section to
clarify this important point.

**To Reviewer #2:**

*This paper presents a detailed study of the magneto-transport of SrIrO₃/SrTiO₃ superlattice.*
*Combined with the magnetoresistivity measurement, X-ray spectroscopy and model calculation,*
*the authors associate the positive magnetoresistivity with the antiferromagnetic spin fluctuation*
*inherent to the Slater-Mott crossover. Although the paper presents some interesting concepts, I*
*could not find any outstanding features in the experimental result. The main result*
*(magnetoresistivity and XMCD) appears to be conventional and to require the no special*
*concept. In addition, there are many open questions about the data analysis and interpretation to*

*support the author's conclusion. I therefore do not recommend publication of this manuscript in*
*Nature Communications.*

**Response:**

We thank the referee for the helpful comments, but would like to ask for reconsideration
of the assessment of our results as being “*conventional and to require the no special concept*”.
Our work presents a novel mechanism for magnetoresistance in the *Slater-Mott crossover regime*
that is fundamentally different from that in conventional magnetic metals and semiconductors.
This mechanism is driven by *a collective charge response to the longitudinal antiferromagnetic*
*fluctuations suppressed by the external field due to strong spin-orbit interaction*. Our
observations highlight the fact that the same MR measurement on different materials can probe
drastically different physical processes and reveal new microscopic mechanisms because of the
rich physics underlying MR.

Specifically, we discovered a large positive MR, in stark contrast to the usually observed
small and negative MR, above the ordering temperature in an AFM insulating/semiconducting
material. We added statement “*In other materials, MR of this magnitude in the paramagnetic*
*state is usually negative and bears an insulator-to-metal phase transition*^{23,24}, *highlighting the*
*unusual combination of robust insulating/semiconducting behavior and large positive MR that is*
*present in the SL.*” in the 1st paragraph on Page 7 to highlight this contrast and our discovery. By
combining our transport measurements with XMCD and theoretical modelling, we
unambiguously unraveled the underlying origin of the observed MR as produced by the *electron-*
*hole paring fluctuations* and correspond to *longitudinal* fluctuations of the AFM order parameter.
This strong spin-charge interplay above the ordering temperature is a key feature of the Slater-
Mott crossover regime. We have elaborated this point in the revised manuscript: “*The local*

magnetic moments arise from the formation of particle-hole pairs, while the antiferromagnetic
(AFM) ordering corresponds to a condensation of these pairs. Correspondingly, the suppression
of the AFM moments can be associated with a reduction in the number of particle-hole pairs.
This process leads to an increase in the number of free particles and holes promoted to electron-
hole continuum above the Mott gap (Fig. 1b).” in the 1st paragraph on Page 3.

What is also very *unconventional* in our heterostructure is that *it is designed to exploit the*
*large spin-orbit coupling that enables a linear coupling between the external uniform magnetic*
*field and the staggered moment.* In other words, the combination of spin-orbit coupling and
crystal field allows to convert the external uniform field into an effective staggered effect that
suppresses the longitudinal AFM fluctuations (i.e. it reduces the number of *unbounded* electron-
hole pairs) inducing a positive MR without the long-range magnetic order. This behavior cannot
be observed in regular antiferromagnets where the staggered field effect is simply absent. We
have further highlighted this point in the revised manuscript by adding “*This novel*
*magnetoresistance originates from a collective charge response to the large longitudinal spin*
*fluctuations under a linear coupling between the external magnetic field and the staggered*
*magnetization enabled by strong spin-orbit interaction.*” in the abstract.

*1. The positive magnetoresistivity is observed nearby the magnetic transition temperature (T_N),*
*but also substantially remains well below T_N (Fig. 2a). On the other hand, the model*
*calculation (Fig. 4d) shows that the magnetoresistivity is observed only nearby T_N . The theory*
*appears to fail to explain the experimental result even in qualitative level. Authors should*
*append further explanation.*

**Response:**

We thank the referee for pointing this out. We would like to first clarify that the primary
goal of this study is to understand the regime above and near T_N , where finite temperature
resistivity is very difficult to model because spatial AFM fluctuations are present. Our model
does not include effects that become significant only when the temperature becomes much lower
than the charge gap, such as domain switching, domain wall, domain pinning, and defects, which
of course always exist in real materials and are extrinsic to the physics we aim to address.
Therefore, the disagreement between the calculation and the experiment well below T_N is not
surprising. We added “Additionally, extrinsic effects that may dominate the conductivity well
below T_N , such as magnetic domains or domain walls^{40,41} cannot be captured by our model.” in
the 1st paragraph on Page 14 to clarify this point.

Specifically, our simulations of the 2D Hubbard model on a finite lattice of 64 x 64 sites
cannot describe the multi-domain physics. Therefore, extrinsic effects, such as domain pinning,
domain wall motion and magnetic hysteresis, which always occur in real macroscopic magnetic
materials, are not considered. These effects become significant only when the temperature is well
below T_N . They are negligible at high temperatures because of the absence of spontaneous long-
range magnetic order. As different magnetic domains start to develop below T_N , the MR is
significantly affected by these extrinsic factors. One possibility, as pointed out by Referee#1, is
the domain wall contributions to the low temperature MR. Another possible effect is that, when
the applied field is smaller than the coercivity field, a portion of the domains will have their
canted magnetization opposite to the applied field, and their AFM order will be subject to an
opposite staggered field and relatively suppressed. Their conductivity will thus be lower than that
of the domains that are parallel to the field. Varying the field would induce MR of the entire

sample by modulating the magnetic domains population and their averaged resistivity. This is
also consistent with our picture of amplitude modulation of the AFM moments. While the above
mentioned extrinsic effects become important at low temperatures, our measurements show that,
as expected, they give a very small contribution to the total MR at temperatures $T > \sim T_N$ (Please
refer to Fig. R1).

Additionally, defects always exist in real macroscopic samples. Defects in
semiconducting materials are known to cause variable-range hopping conduction at temperatures
much smaller than the charge gap. This effect is not considered in our model and will lead to
deviation from thermal activation conduction at low temperatures. Indeed, at temperatures well
below T_N , the experimental resistivity deviates from the calculated temperature dependent
resistivity based on thermal activation. Such deviation from thermal activation conduction at low
temperatures toward variable-range hopping is quite common in oxide materials. This simple
fact indicates that the calculated MR is not expected to fully reproduce the experimental one in
that temperature range.

Nevertheless, the presence of extrinsic effects that become significant well below T_N is
irrelevant for capturing the physics behind the rapidly increasing positive MR when cooling
down to T_N . As shown in Fig. 4d, this feature is indeed well described by our theoretical model
qualitatively. The consistency between the experimental data and theoretical model enables one
to unambiguously uncover the origin of the anomalous MR as the strong coupling between the
particle-hole pair strength and longitudinal spin fluctuations, which is still finite even at
temperatures above T_N . Thus, our work focuses on these fluctuations above and near T_N . To
further clarify the above points, we added discussions “We found that these spin-charge
fluctuations can be controlled with an external magnetic field, which couples linearly to the

staggered magnetization (Fig. 1c) due to the strong spin-orbit interaction (SOI), and induces a
large positive magnetoresistance (MR) above T_N . The observed effects are well reproduced by
our calculation, which captures the spatial AFM fluctuations of the paramagnetic state.” in the
2nd paragraph on Page 4, and “The calculated MR at temperatures above T_N is also positive and
it reaches a maximum at the AFM transition in agreement with the experimental observation
(Fig. 2a). In stark contrast, M^{st} and ρ remain almost unchanged under the effect of an out-of-
plane field due to the lack of the staggered field effect that is expected from Eq. (2).” in the 1st
paragraph on Page 14.

2. The reason why the XMCD can detect the antiferromagnetic order parameter (staggered
susceptibility) is not clear. In general, XMCD is scaled to the uniform (ferromagnetic)
susceptibility. In this sense, the resonant magnetic scattering at (0.5, 0.5, 2) would be related to
the antiferromagnetic order parameter. It is not clear why authors associate the
magnetoresistivity with the XMCD rather than the resonant magnetic reflection to discuss the
antiferromagnetic order parameter.

**Response:**

The referee is correct that XMCD characterizes the net magnetization, please see the
updated discussion “XMCD measures the uniform magnetization, which at zero magnetic field
characterizes the canted component of the spontaneous AFM order parameter (OP) as can be
seen from its similar T-dependence to the AFM Bragg peak (Fig. 1f). The field-induced XMCD
variation is thus proportional to the uniform susceptibility χ , which indeed displays a clear
maximum around T_N (Fig. 2b).” in the 1st paragraph on Page 8. We utilized XMCD here to probe

the AFM order by exploiting the spin-canting nature of the net magnetization. For a canted
antiferromagnet, the net magnetization \mathbf{M} is related to the staggered magnetic moment \mathbf{M}^{st} as \mathbf{M}
$= \mathbf{M}^{st} \cdot \sin\phi$, where ϕ is the spin canting angle. This allows XMCD to probe the AFM order as
demonstrated in Fig. 1f. ϕ is considered to be a constant for the field that we applied (please see
the response below for comment #3). When an external magnetic field \mathbf{H} is applied along the
direction of \mathbf{M} , it couples to \mathbf{M}^{st} as an effective staggered field \mathbf{H}^{st} with the strength of $|\mathbf{H}| \cdot \sin\phi$.
Therefore, the uniform magnetic susceptibility $\chi = d\mathbf{M}/d\mathbf{H}$ is proportional to the staggered
susceptibility $\chi^{st} = d\mathbf{M}^{st}/d\mathbf{H}^{st}$ with a factor of $\sin^2\phi$ [*Phys. Rev. Lett.* **106**, 136402 (2011); *Phys.*
*Rev. B* **94**, 224402 (2016)].

We certainly agree with the referee that ideally one would like to monitor the AFM order
parameter by directly measuring the magnetic Bragg peak intensity. However, the maximum
magnetic field of the diffractometer at 6IDB beamline in Argonne National Lab was only 0.5 T,
much smaller than that used in the MR measurement. Combined with the fact that thin film
samples have relatively weak signals due to the small sample volume, capturing the field-
induced response at T_N with 0.5 T to sufficient statistics is an unrealistic task within the duration
of the beamtime. On the other hand, the XMCD experiment can be performed under a much
higher magnetic field ~ 5 T (4IDD beamline in Argonne National Lab) and thus is more feasible
as we demonstrated in the manuscript.

*3. Authors assume that the canting angle phi is unchanged even nearby antiferromagnetic*
*transition temperature. The reason is not clear. The uniaxial anisotropy is usually well defined*
*when the magnetic order parameter is sufficiently large (longitudinal magnetic fluctuation is*

negligible). Moreover, regarding this point, it is not comprehensible that the magnetic field
induces a finite antiferromagnetic order parameter above T_N .

**Response:**

We should distinguish between two different concepts: the parameter φ defined in Eq. (1),
and the canting angle ϕ determined by the relative angle of neighboring ordered spins. In the
revised manuscript, we will use the two different symbols to avoid confusion. We next explain
the two parameters separately.

**a)** In Eq. (2), the last term reveals that magnetic field couples linearly to the staggered
magnetization (order parameter) in a twisted Hubbard model, consistent with previous
report [*Phys. Rev. Lett.* **106**, 136402 (2011)]. The ultimate mechanism for this coupling is
the spin-dependent hopping that exists in the global spin frame. The factor $\sin\varphi$ explains
the underlying charge effect because φ characterizes the spin-dependent hopping phase
factor in Eq. (1). It has been shown before that φ in square-lattice iridates is determined
by a combination of SOI and octahedral distortion [*Phys. Rev. Lett.* **102**, 017205 (2009);
*Phys. Rev. Lett.* **106**, 136402 (2011)]. Since both parameters are temperature-independent,
φ is a constant. Therefore, the linear coupling between magnetic field and staggered
magnetization can persist to any temperature, enabling presence of finite staggered
magnetization under uniform magnetic fields even at temperatures above T_N .

**b)** In zero field, the experimental manifestation of φ is the spin canting angle ϕ . This can
also be understood by using the model Hamiltonian in the rotated frame (Eq. (2)). In zero
field, it gives an antiferromagnetic ground state, implying that the neighboring spins in
the rotated frame align in opposite directions. Going back from the rotated frame to the

global frame requires a spin rotation about the z -axis by an angle φ in one sublattice and
by an angle $-\varphi$ in the other sublattice. In other word, the canting angle ϕ equals to φ in
zero field. Given that φ is a constant, the canting angle ϕ is temperature-independent as
well (in zero field). With a finite field, the two objects (ϕ and φ) are no longer identical
strictly speaking. Their ratio depends on applied magnetic fields, and the uniform and
staggered g -factors. The relation has been explicitly deduced by using spin-wave theories
in the *Methods section* in *Nat. Commun.* **6**, 7306 (2015). After inputting relative
parameters of our superlattice obtained from a recent resonant inelastic x-ray scattering
experiment [*Sci. Rep.* **9**, 4263 (2019)], one can estimate that a 14 T in-plane magnetic
field changes ϕ by $\sim 0.07^\circ$. This is more than two orders of magnitude smaller than the
zero-field value, $\sim 10^\circ$. Therefore, we conclude that ϕ can be approximated as a
temperature independent constant in the current study. This justifies the utilization of
XMCD to characterize the staggered susceptibility as mentioned in the response to
comment #2.

To emphasize the above points, we have added another figure showing the spin-
dependent hopping in Fig. 3 and additional discussions, like “The zero-field canting angle is
temperature independent and it is determined by the magnitude of the $SOI^{10,21}$.” in the 1st
paragraph on Page 5, and “Interestingly, since the canting angle $\phi \sim 10^\circ$ is determined by a
combination of the lattice distortion and the strong $SOI^{10,21}$, and is practically unchanged for
these field values²⁶, the measured uniform susceptibility χ becomes proportional to the staggered
susceptibility χ^{st} near T_N with a proportionality factor $\sin^2\phi \sim 0.03^{21,27}$.” in the 1st paragraph on
Page 9, and “Unlike the usual spin-half Hubbard model, this phase factor renders complex
hopping integrals for different spins due to the spin-orbit-entangled $J_{\text{eff}} = 1/2$ wavefunctions. It is

important to note that the in-plane spin canting in the AFM ground state is ultimately driven by
this spin-dependent hopping and the angle ϕ of the phase factor determines the canting angle ϕ
at zero field^{35,36}. At finite fields, it determines the ratio of the uniform susceptibility and the
staggered susceptibility. Therefore, the spin-dependent hopping allows the external magnetic
field to couple linearly with the staggered magnetization through the uniform component at any
temperature.” in the 2nd paragraph on Page 11.

4. In the discussion part, authors call this magnetoresistivity as “magnetoelectric effect”.
However, conventionally, the magnetoresistivity is not termed as magnetoelectric effect.

**Response:**

Thanks for pointing this out. We used “magneto-electronic effect” through the manuscript.

**To Reviewer #3:**

The authors report a magnetoresistance (MR) measurement in the antiferromagnetic Mott
insulator superlattice SrIrO₃/SrTiO₃. Distinct from the negative MR found in other Mott
insulators, the MR in the SrIrO₃/SrTiO₃ superlattice is positive which shows a maximum at the
Neel temperature. The positive MR observed is interpreted with the magnetic field suppression of
the spin fluctuation and a theoretical calculation is included in the manuscript. I hope the
authors can address the following issues, which will help me to judge whether the present work
deserves the publication in Nature Communications.

**Response:**

We value the referee's inputs and constructive comments from reviewing our work.

The following are detailed response to each point.

*1. The magnitude of the MR in reported in this work is about 1%/T which is strongest near the*
*Neel temperature. Have to say that such a MR magnitude is not really "large" compared to the*
*MR in the TMR and GMR device which are applicable. However, MR with small magnitude can*
*still be indispensable for the characterization of material properties. For example: the recently*
*discovered spin Hall magnetoresistance [H. Nakayama et al., PRL 110, 206601 (2013)][D. Hou*
*et al., PRL 118, 147202 (2017)], can be used to probe the spin diffusion length, spin Hall angle,*
*the spin mixing conductance and even the interface magnetic moment directions in spite of the*
*0.01% MR magnitude. So, are the authors able to evaluate some parameters in this*
*SrIrO3/SrTiO3 superlattice? This is also an important reference for this reviewer to judge the*
*novelty of the present work.*

**Response:**

We thank the referee for this good suggestion. We completely agree that MR
characterization is indispensable. It leads to significant results in our work that reveals a new
mechanism for magnetoelectronic effects that is fundamentally different to conventional
magnetoresistance and TMR/CMR. It is enabled by controlling the collective charge response to
the longitudinal spin fluctuations within the Slater-Mott crossover regime. In other words, the
anomalous MR effect is an electronic probe of the longitudinal AFM fluctuations with the
magnitude determined by the staggered susceptibility rather than the AFM order parameter.

Following the referee’s suggestion, we further evaluate the observed magnetoelectronic
response in the paramagnetic insulating state, which is about $\sim 1\%/T$ near T_N . Based on the
thermal activation behavior of resistivity above T_N , we estimate the field-induced enhancement
of the activation energy gap to be 0.12 meV by every 0.02×10^{-3} meV Zeeman energy. This
renders an amplification coefficient of ~ 6000 in energy scale from the spin to the charge degree
of freedom, which is huge and is a result of the combination of a very large staggered
susceptibility ($0.05 \text{ emu} \cdot \text{mol}^{-1} \cdot \text{Oe}^{-1}$) of the insulating/semiconducting Slater-Mott crossover
regime and a large spin-dependent phase factor ($\sim 10^0$) of the hopping produced by the strong
spin-orbit coupling. We would like to emphasize that our superlattices were designed to combine
these ingredients in order to enable such amplification.

We have added related discussions to further clarify the above points, “Therefore, based
on the extracted χ from XMCD and the thermally activated resistivity above T_N , we estimate the
MR of $\sim 1\%/T$ to be corresponding ~ 0.12 meV enhancement of the activation energy by every
$\sim 0.02 \times 10^{-3}$ meV of the Zeeman energy $\mu_0 \cdot \chi \cdot H^2$, i.e., a response coefficient of ~ 6000 in energy
scale. In other words, the effect of the external magnetic field is amplified by more than three
orders of magnitude in the electronic response due to the strong interplay between spin and
charge.” in the 1st paragraph on Page 8, and “Note that, although the magnitude of the observed
anomalous MR is smaller than the GMR⁴⁷ and CMR effects²⁵ of magnetic metals, novel MR
effects often indicate a new physical mechanism, like the one present in the recently discovered
spin-Hall MR effect^{48,49}. In our case, the sensitivity of MR to the longitudinal spin fluctuations
provides an efficient electronic probe of the usually elusive staggered susceptibility of Mott-type
insulating materials.” in the discussion section and added ref. 48 [PRL 110, 206601 (2013)] and
ref. 49 [PRL 118, 147202 (2017)].

2. I wonder whether the authors can make some argument about other systems which probably
show a positive MR due to the same mechanism. If a phenomenon can be observed in a variety of
materials, it is very likely to be important and of general interest.

**Response:**

The key to enabling the observed positive MR and/or related electronic responses to
external field lies in the combination of spin-orbit coupling in AFM insulating systems for
modulating the spin-charge interplay in the *Slater-Mott crossover* regime. To the best of our
knowledge, this phenomenon has not been reported in other systems. Although many materials
exhibit Slater-Mott crossover physics, only a few of them have a spin-dependent hopping driven
by strong spin-orbit coupling. Indeed, strongly spin-orbit-coupled correlated materials, such as
*5d* transition metal oxides, became the focus of interest only a few years ago. However, most
studies focused their attention on the ground state properties rather than the fluctuations in the
paramagnetic state. We believe that our results will motivate the community to look for similar
materials and effects, and the superlattice in this work is only the beginning of this search. To
further demonstrate this point, we have designed another artificial layered iridate,
[(SrIrO₃)₂/(SrTiO₃)₁] on SrTiO₃ (001) substrate, which is similar to the sister [(SrIrO₃)₁/(SrTiO₃)₁]
superlattice, featuring an intermediate coupling strength, a canted AFM ground state and a
resistance anomaly around the Neel temperature [*Phys. Rev. Lett.* **114**, 247209 (2015); *Phys. Rev.*
*Lett.* **119**, 027204 (2017); *Sci. Rep.* **9**, 4263 (2019)]. As shown in Fig. R2, the positive anomalous
MR is observed as well in this superlattice around the Neel temperature. As pointed out in the
reply to Referee #1, it is important to avoid any internal effect that competes with the external
field for engaging with the spin-dependent hopping.

In summary, although there is currently no example of a bulk material combining all the
necessary ingredients for producing the reported MR effect, our study demonstrates that,
knowing the guiding principles, it is possible to design and build heterostructures that optimize
the effect. This is an example of materials by design that opens a new dimension for the study
and exploitation of correlated electron phenomena. Moreover, the same guiding principles can be
applied to the search of new bulk materials that should exhibit this MR effect.

3. In line 54, it reads "...to control and practical Mott materials," the "and" should be "of",
right?

**Response:**

Thanks for this point. We rewrote the sentence "*It is however difficult to detect and*
*exploit this spin-charge fluctuations because AFM order is hard to control and practical Mott*
*materials, like the parent phase of high-Tc cuprates, are often deep inside the Mott regime where*
*the charge degree of freedom is frozen and charge transport is diminished.*" as "*It is however*
*difficult to detect and exploit this spin-charge fluctuations because AFM order is hard to control.*
*Moreover, practical Mott materials, like the parent compounds of high-Tc cuprates, are often*
*deep inside the Mott regime where the charge degree of freedom is frozen and charge transport*
*is diminished.*"

4. I suggest the authors to specify the orientation of the STO substrates (is it STO(001)?).

**Response:**

According to this suggestion, we have added the orientation of the STO substrate in the
Method section.

*5. Does the direction of magnetic field in the a-b plane matter for the MR result? And please*
*specify which direction the field is applied during the measurement.*

**Response:**

This is a good point. The magnitude of the MR is indeed sensitive to the direction of the
applied magnetic field within the *ab*-plane. However, the contribution from this anisotropic
magnetoresistance effect is relatively small and does not alter the overall temperature
dependence of MR. As shown in Fig. R3, the MRs measured with the magnetic field along the
[100] and [110] directions and the current along the [100] direction differ by ~2%, while their
average is around 9% near 150 K. Their difference below T_N is much more significant than that
above T_N . This indicates that the difference between the two directions is due to in-plane
anisotropy of the square lattice, the energy of which is expected to be larger as the order
parameter increases at lower temperatures. We have added this data in the supplement Material
and specified the field direction in the Method part.

**Figure R3.** Temperature dependent MR with an in-plane magnetic field applied along the [110]
 (red) or [100] (black) direction.

6. For the argument made in line 124-126, please add more reference to support it, otherwise it
 is very hard to judge for general audience.

**Response:**

To avoid confusion, we changed the sentence as “In other materials, MR of this
 magnitude in the paramagnetic state is usually negative and bears an insulator-to-metal phase
 transition^{24,25}, highlighting the unusual combination of robust insulating/semiconducting
 behavior and large positive MR that is present in the SL.” and added ref. 24 [*Curr. Opin. Solid*
 *State Mater. Sci.* **2**, 244-251 (1997)] and ref. 25 [*J. Phys.: Condens. Matter* **9**, 8171-8199 (1997)].

7. I wonder whether it is necessary to emphasize the “magnetic field is equivalent to a staggered
field”, because this argument is too general and it is essentially the vector addition.

**Response:**

We agree with the referee that the effective staggered field is a general idea. Nevertheless,
considering the response to the Referee#2’s comments and the general readership of *Nature*
*communication*, we feel it is necessary to provide a brief introduction of this idea. In fact, as
discussed in the response to Referee#2 and the revised manuscript, it is important to note that the
staggered field effect persists in the paramagnetic state without the spontaneous magnetic order,
as it is evident from the last term of Eq. (2).

We hope the above responses will relieve the referee’s concern and deliver a positive
impression of this work.

**Next, we list all other major changes which were not mentioned in the above responses.**

**1) Abstract**

Add “Its size is particularly large in the high-temperature insulating paramagnetic phase near
the Néel transition.”

**Text**

**2) Line 51.**

Delete “*Correspondingly, it follows that magnitude variation or elimination of the AFM*

*moment necessarily evokes charge delocalization or excitations to the electron-hole*
*continuum, and vice versa (Fig. 1b)."*

**3) Line 63.**

Replace "*On the other hand, the significantly reduced Coulomb interaction of the 5d*
*electrons also leads to interpretation of the AFM insulating behavior as a Slater insulator,*
*which is the solution of the Hubbard Hamiltonian at the weak coupling limit."* with "On the
other hand, the significantly reduced Coulomb interaction and larger extension of the 5d
orbitals shifts these materials toward the Slater regime corresponding to the solution of the
half-filled Hubbard Hamiltonian in the weak-coupling limit."

**4) Line 79.**

Delete "*Specifically, longitudinal spin fluctuations can revive in the "normal state" of the*
*AFM order without completely annihilating the magnetic moments. The corresponding*
*fluctuations of the electron-hole pairing will thaw without closing the Mott gap and sharply*
*respond to suppression of the longitudinal spin fluctuations."*

**5) Line 112.**

Replace "*shows anomalous T-dependence beyond the Mott-Heisenberg scheme"* with
"reveals an anomalous T-dependence that cannot be explained within the Mott-Heisenberg
scheme".

**6) Line 124.**

Replace "*considering that the Zeeman energy of the external field is at least two orders of*
*magnitude smaller than T_N "* with "considering the absence of spontaneous long-range
magnetic order above T_N ".

**7) Line 137.**

Replace "*XMCD characterizes the AFM order parameter (OP) through the canted*
*component as can be seen from its similar T-dependence to the AFM peak (Fig. 1f). Since*
*the canting angle ϕ is practically unchanged for these field values, we can consider the*

*field-induced XMCD variation to be directly proportional to the response of the AFM OP,*
*i.e., the staggered susceptibility. The XMCD variation indeed shows a susceptibility-like*
*behavior as it becomes observable below 180 K, reaches a maximum around T_N , and drops*
*upon further cooling (Fig. 2b).” with “XMCD measures the uniform magnetization, which at*
*zero magnetic field characterizes the canted component of the AFM order parameter (OP) as*
*can be seen from its similar T -dependence to the AFM Bragg peak (Fig. 1f). The field-*
*induced XMCD variation is thus proportional to the uniform susceptibility χ , which indeed*
*displays a clear maximum around T_N (Fig. 2b).”.*

**8) Line 151.**

Replace “*by inducing a finite AFM OP above T_N via the spin canting*” with “near T_N via
the large staggered susceptibility.”.

**9) Line 152.**

Add “In other words, the MR above T_N is the charge response to the large relative
increase of the staggered magnetization induced by the external field.”.

**10) Line 155.**

Add “and the uniform susceptibility is not sensitive to the staggered susceptibility”.

**11) Line 160.**

Replace “*It is known that the charge gap of a single-band Hubbard system results from*
*the AFM order-induced band reconstruction in the weak-coupling (BCS) limit and is directly*
*proportional to the staggered magnetization M^{st} ” with “The large anomalous MR in the*
*paramagnetic phase is clearly incompatible with a Mott-Heisenberg regime where charge degrees*
*of freedom are basically frozen because of a charge gap that is much larger than the hopping*
*amplitude. In the opposite weak-coupling limit or Slater regime¹⁴, the charge gap arises from the*
*band reconstruction induced by the AFM ordering and it is directly proportional to the staggered*
*magnetization M^{st} .”*

**12) Line 167.**

Replace “*which is incompatible with the observation*” with “and a field-induced gap
much smaller than T_N does not necessarily affects the resistivity, which clearly would not
account for our observations.”.

**13) Line 174.**

Replace “*particle-hole pairs*” with “magnetic moments”.

**14) Line 198.**

Delete “*It is responsible for the in-plane spin-canting and leads to a Dzyaloshinskii-*
*Moriya interaction in the large- U/t limit^{31,32}”.*

**15) Line 207**

Rewrite the Hamiltonian in an explicit way.

**16) Line 215.**

Replace “*an externally induced effective staggered field effect*
*as $\sin\phi \mathbf{h}_{st}^\perp \cdot \sum_j \tilde{\mathbf{s}}_j^\perp \exp(i\mathbf{Q} \cdot \mathbf{r}_j)$, where \mathbf{h}_{st}^\perp and $\tilde{\mathbf{s}}_j^\perp$ are transformed in-plane field and spin,*
*respectively.” with “the Hubbard model with the usual spin-independent hopping and a linear
coupling between the external field and the staggered magnetization scaled by $\sin\phi$.”.*

**17) Line 225.**

Add “after including the easy-plane anisotropy that accounts for the Hund’s coupling
(Methods).”.

**18) Line 248.**

Replace “*This analysis uncovers the role of SOI, which allows a uniform field*
*to exploit the large longitudinal staggered susceptibility of the Slater-Mott crossover regime to*
*create a spin-alternating potential that ties up the particle-hole pairs (Fig. 1c), giving rise to a*
*large positive anomalous MR*” with “This analysis uncovers the role of the SOI, which enables a
linear coupling between the uniform field and the staggered magnetization and therefore a
relatively large positive anomalous MR due to the large longitudinal staggered susceptibility of
the Slater-Mott crossover regime.”.

**19) Line 304.**

Add “which features a high magnetic field strength of 6 T.”.

**Subtitle**

**20)** Replace “*AFM-spin-tunable effective carrier density within Slater-Mott crossover regime.*”
with “Modelling the anomalous magnetoelectronic response in Slater-Mott crossover”.

regime".

**Figures**

**21) Figure 2**

Replot the data as uniform susceptibility in Fig. 2(b).

**22) Figure 4**

Correct the typo of the y-axis label of Fig. 4(b).

**Figure caption**

**23) Figure 2**

Replace "*The in-plane field-induced difference of the XMCD between 5 and 0 T versus*
*temperature.*" with "In-plane uniform susceptibility χ extracted from the in-plane field-
induced XMCD difference (Supplementary 3).".

**24) Figure 3**

Add "Fig. 3 Charge hopping (spin-up channel) in different spin frames. In the global spin
frame (left panel), charge hopping bears an alternating phase factor when circling around the
square lattice. This phase factor is gauged away in the rotated local spin frame (right panel),
leading to an isotropic Hubbard model^{21,27}. Charge hopping in the spin-down channel can be
obtained after applying the time-reversal symmetry operation. The annihilation operators
$\tilde{c}_{j\alpha\beta}$ in the local frame are transformed from $c_{j\alpha\beta}$ in the global frame according to the shown
transformation."

**25) Acknowledgments:**

Add "This work used resources of the Compute and Data Environment for Science
(CADES) at the Oak Ridge National Laboratory, which is supported by the Office of
Science of the U.S. Department of Energy under Contract No. DE-AC05-00OR22725. This
research used resources of the Oak Ridge Leadership Computing Facility, which is a DOE
Office of Science User Facility supported under Contract DE-AC05-00OR22725".

**References**

**26) Add references:**

**24** Bishop, A. R. & Roder, H. Theory of colossal magnetoresistance. *Curr. Opin. Solid State*
*Mater. Sci.* **2**, 244-251 (1997).

**25** Ramirez, A. P. Colossal magnetoresistance. *J. Phys.: Condens. Matter* **9**, 8171-8199
(1997).

**27** Takayama, T., Matsumoto, A., Jackeli, G. & Takagi, H. Model analysis of magnetic
susceptibility of Sr₂IrO₄: A two-dimensional $J_{\text{eff}} = 1/2$ Heisenberg system with competing
interlayer couplings. *Phys. Rev. B* **94**, 224420 (2016).

**40** Seidel, J. *et al.* Conduction at domain walls in oxide multiferroics. *Nat. Mater.* **8**, 229
(2009).

**41** Choi, T. *et al.* Insulating interlocked ferroelectric and structural antiphase domain walls in
multiferroic YMnO₃. *Nat. Mater.* **9**, 253 (2010).

**42** Meyers, D. *et al.* Magnetism in iridate heterostructures leveraged by structural
distortions. *Sci. Rep.* **9**, 4263 (2019).

**43** Ge, M. *et al.* Lattice-driven magnetoresistivity and metal-insulator transition in single-
layered iridates. *Phys. Rev. B* **84**, 100402(R) (2011).

**44** Wang, C. *et al.* Anisotropic Magnetoresistance in Antiferromagnetic Sr₂IrO₄. *Phys. Rev. X*
**4**, 041034 (2014).

**45** Lee, N. *et al.* Antiferromagnet-Based Spintronic Functionality by Controlling Isospin
Domains in a Layered Perovskite Iridate. *Adv. Mater.* **30**, 1805564 (2018).

**46** Lu, C. *et al.* Revealing Controllable Anisotropic Magnetoresistance in Spin–Orbit
Coupled Antiferromagnet Sr₂IrO₄. *Adv. Funct. Mater.* **28**, 1706589 (2018).

**47** Baibich, M. N. *et al.* Giant Magnetoresistance of (001)Fe/(001)Cr Magnetic
Superlattices. *Phys. Rev. Lett.* **61**, 2472-2475 (1988).

**48** Nakayama, H. *et al.* Spin Hall Magnetoresistance Induced by a Nonequilibrium
Proximity Effect. *Phys. Rev. Lett.* **110**, 206601 (2013).

**49** Hou, D. *et al.* Tunable Sign Change of Spin Hall Magnetoresistance in Pt/NiO/YIG
Structures. *Phys. Rev. Lett.* **118**, 147202 (2017).

All the in-text citations are also updated.

**In addition, all the corrections are marked in red in the revised paper.**

Reviewers' Comments:

Reviewer #1:

Remarks to the Author:

The authors have made satisfactory changes to the manuscript in response to my comments, and this can be accepted for publication.

Reviewer #2:

Remarks to the Author:

Authors have substantially addressed this reviewer's comments and the manuscript has been improved. However, it is still not clear why the data (experiment and theory) supports the proposed scenario. While the authors have provided a thorough response, there are several points that the authors have not sufficiently addressed to fully satisfy the claim.

1) The experiment shows that the substantial magnetoresistivity is observed above and below T_N . Authors present that the reason of discrepancy between the model calculation and experiment below T_N is extrinsic effects such as the domain wall pinning, motion and defects. This is just a speculation and evidence is not shown. Objectively, the model calculation is only consistent with limited temperature range above T_N . This rather suggests that the proposed model is not sufficient to explain the experimental results.

2) The main result (magnetoresistivity and XMCD) appears to be ordinary and there is no surprise. As authors propose in the response, there are many possible mechanisms such as the domain pinning, domain wall motion, and defects, which can induce the similar positive magnetoresistivity. In addition, the reason why these mechanisms or their extension cannot be the origin of the observed results remains to be unclear.

Reviewer #3:

Remarks to the Author:

The authors have improved the paper according to my comment and suggestions, and all the issues from me are properly addressed. I also notice that the 2nd reviewer has some concern, and I think it is also important for the authors to further clarify some issues and convince the 2nd reviewer in this round. So I would like to see the comment from the 2nd reviewer in the next round and make the final decision.

Reviewer #4:

Remarks to the Author:

The manuscript has been gone through a first-round reviewing by three very competent referees. The reply by the authors was done adequately and professionally. A positive magnetoresistance observed (mainly) in the vicinity of magnetic transition temperature on the artificial superlattice insulator is interesting. The physics proposed behind the positive MR could have an impact in the field of strongly correlated electrons with strong spin-orbit coupling. The data quality in the measurements with a suite of probes is good. The manuscript could be eventually acceptable by Nature Communications; but it must go through a major overhaul; perhaps by adding additional experimental data.

The concept of Slater-Mott crossover is not clear and may even be misleading. A Mott transition refers a metal-insulator transition due to electron correlations; the magnetic transition which occurs at a low temperature (in some cases) does not influence transport properties dramatically.

The Slater transition on the other occurs in a band electron system where a change of the translational symmetry at $T < T_N$ introduces a gap at the Brillouin zone boundary. However, I do agree that a maximum T_N should occur at crossover of localized to itinerant electronic behavior as shown in Fig.4 (a).

Fig.1 (abc) is not helpful at all in the introduction for understanding the work carried out in this manuscript and it should be removed. "Spin ordering is to order the hole-electron pair" is simply wrong.

The submission package lacks a thorough description of the sample preparation and characterization, for example, the monitoring of 1:1 SIO/STO layers along the c axis. How many layers of the superlattice were used in the sample for the MR measurement? In the superlattice, the Ir-Ir separation along the c axis is as large as 8 Å, whereas it is about 7 Å in Sr₂IrO₄. Perhaps, this difference can be used to justify a lower T_N in the superlattice. The key point is that the canted spin points to the + -b axis alternately along the c axis, so that the net moment vanishes in Sr₂IrO₄. In contrast, they are on the same direction in the superlattice. The magnetization measurements along a,b,c are helpful to establish the magnetic structure (given that there is no neutron diffraction to confirm the magnetic structure). What is the coercive force of this canted iridate superlattice? It is normally pretty large. In case it falls into the field range of a PPMS, the MR measurement in a loop of scanning magnetic field could be helpful to answer the question of domain boundary scattering by one of the previous reviewers. What is the resistivity of the superlattice? Do the activated resistance in Fig.1(e) and magnetic property depend on the number of layers? Is it possible to measure the MR on a sample with single layer SIO. Ferromagnetic Mott insulator (very few) would show a large negative MR like in the underdoped La_{1-x}CaMnO₃. On the other hand, metals show a positive MR by following the Kohler's rule. If the resistivity in the superlattice is low (perhaps a good conductor at $T > T_N$), the physics of MR may be simply as in the Kohler's rule. As a matter of fact, the MR(H) at 175 K in Fig.2 (c) can be well described by the Kohler's rule. Why do the authors fit the MR(H) in this way and find if it makes sense. In the vicinity of T_N , a large MR effect has been interpreted by suppressing the spin longitudinal fluctuations under magnetic field. Since a dramatic increase of resistance as temperature decreases through T_N , the superlattice looks like a Slater insulator. Can the authors derive a gap (by using the R(T) at lower temperatures and separate the two contributions of activated R(T) due to the gap and the extra resistance due to the spin fluctuations in the resistivity change near T_N ?

Fig.1 (d) shows the magnetic structure schematically, whereas the spin structure in Fig.3 is essentially ferromagnetic. Actually, Fig. 3 is not meaningful for understanding the results presented. The authors have introduced key concepts that scattering over places through the text. A clear picture of physics behind the results, such as why the crossover is needed for providing a channel connecting spin and the transport, why strong SOC is needed to give rise a canting (perhaps through the particular superlattice structure) and so on, should be given in the abstract or the introduction.

Moreover, more references about SIO are necessary to cover the knowledge base about this material, for example the layer-dependence of transport behavior (PRL 119, 256403) and to show that the perovskite SIO is close to the magnetic instability.

Reviewer #5:

Remarks to the Author:

This paper reports on the coupling between staggered magnetic moment and external magnetic field in a SrIrO₃/SrTiO₃ superlattice, which shows positive in-plane magnetoresistance. Combined with model calculation, the authors revealed that the positive magnetoresistance originates from the phase factor in the hopping term given by SOI.

Apart from detailed preciseness of the descriptions, two main factors are central debates whether this manuscript deserves Nature Communications, which are positive in-plane magnetoresistance near T_N in an insulating regime and the mechanism of the positive magnetoresistance. In my opinion, the MR and the mechanism found by the authors may be new. However, I am not quite positive to recommend this paper for Nature Communications, given that Nature Communications requires "novelty".

In my opinion, a proper understanding of MR may be scientifically important but just proposing a new mechanism is not very "novel" unless it is very dramatic (e.g. huge MR or in an extremely small magnetic scale) or it gives important physical parameters (e.g. strength of SOI or scattering times in the case of (anti)-weak localization). Nowadays, there are many mechanisms to give various MRs in various conducting regimes from the metallic to hopping conduction. For example, positive magnetoresistance in a similar magnitude is also proposed for the hopping regime [L. Essaleh et al., Phys. Rev. B 52, 7798 (1995); W. Jiang et al., Phys. Rev. B 49, 690 (1994)] although the mechanism may not be the same as the present case.

Having said this, the manuscript itself is relatively sound (except the concerns shown below) and should be published somewhere. At least, this paper proposes a possible new mechanism of positive MR involving SOI in the hopping term. However, this manuscript may not be suitable for Nature Communications in my opinion.

Other than the above concern, the following points should be considered.

1. The structure of the sample is still not clear. First of all, the number of periods is not shown. Second, the authors should show experimental data of the sample, for example, by XRD or TEM. $(\text{SrIrO}_3)_1/(\text{SrTiO}_3)_1$ superlattice is not a trivial structure to fabricate and experimental evidence is necessary to show the sample used in this study is really $(\text{SrIrO}_3)_1/(\text{SrTiO}_3)_1$ superlattice.
2. As pointed out by the other referee, the magnitude of MR (1%/T) is not large, even small. To advertise the magnitude of MR, the authors compared the activation energy and Zeeman energy. However, it is quite rare that Zeeman energy directly couples with the activation energy of electrical conduction. In my opinion, this comparison and the response coefficient of ~ 6000 is meaningless.

Reviewer #1 (Remarks to the Author):

The authors have made satisfactory changes to the manuscript in response to my comments, and this can be accepted for publication.

Response:

We thank the referee for reviewing our work. We highly appreciated the positive recommendation and all the constructive criticism.

Reviewer #2 (Remarks to the Author):

Authors have substantially addressed this reviewer's comments and the manuscript has been improved. However, it is still not clear why the data (experiment and theory) supports the proposed scenario. While the authors have provided a thorough response, there are several points that the authors have not sufficiently addressed to fully satisfy the claim.

Response:

We thank the referee for acknowledging the improvement from the last response. We address the two remaining comments one by one in the following.

1) *The experiment shows that the substantial magnetoresistivity is observed above and below T_N . Authors present that the reason of discrepancy between the model calculation and experiment below T_N is extrinsic effects such as the domain wall pinning, motion and defects. This is just a speculation and evidence is not shown. Objectively, the model calculation is only consistent with limited temperature range above T_N . This rather suggests that the proposed model is not sufficient to explain the experimental results.*

Response:

We agree with the referee that our theory works at temperatures near and above T_N , but shows a discrepancy with experimental observations at temperatures well below T_N . We also understand that, ideally, one would expect a theoretical model that holds for temperatures higher than T_N as

well as temperatures lower T_N . However, to the best of our knowledge, there is no effective model which works to arbitrary low or high temperatures for a given material. For instance, the Curie–Weiss law is known to fail in the vicinity of a magnetic transition due to critical fluctuations, but this shortcoming does not diminish its significance. In our case, since the MR already emerges above T_N and is maximized near T_N , we set the main goal of the model calculation to be capturing the critical spin fluctuations underlying this anomalous temperature dependence. We would like to point out that such critical magnetic behavior is difficult to be modeled, because one has to consider both longitudinal (magnitude change of spin) and transverse spin fluctuations (direction change of spin) on the equal footing while they vary by orders of magnitude even within a small temperature window in this critical regime. Note that, since there is no free parameter in our model and all the inputs are profound values of iridates, the qualitative consistency between the experimental observation and the theoretical prediction in the temperature regime of interest is actually quite remarkable.

We did not include magnetic domain effects in our theoretical model, since there is no magnetic long-range ordering when temperature is higher than T_N . Domain-related effects are usually negligible near T_N due to the dominant role of critical fluctuations. On the other hand, the MR data below T_N included in the last reply letter clearly shows a feature related to magnetic hysteresis (please refer to the 1st response to the referee#1 and Fig. S4 in the supplementary material). This is an evidence of extrinsic contributions from domain effects. In other word, the discrepancy between experiment and theory at temperatures well below T_N is somewhat as expected. The hysteresis is quickly diminished as expected when the temperature increases toward T_N . Therefore, excluding this extrinsic effect does not prevent us from capturing the intrinsic physics.

2) The main result (magnetoresistivity and XMCD) appears to be ordinary and there is no surprise. As authors propose in the response, there are many possible mechanisms such as the domain pinning, domain wall motion, and defects, which can induce the similar positive magnetoresistivity. In addition, the reason why these mechanisms or their extension cannot be the origin of the observed results remains to be unclear.

Response:

We would like to emphasize that a key finding of the current work is a positive and large anomalous MR, especially at temperatures above T_N , which as discussed in the manuscript is a unique phenomenon in the paramagnetic phase of a Hubbard system within the Slater-Mott crossover regime. As explained in the last reply, mechanisms based on magnetic domains and their related effects do not apply to the paramagnetic state simply because there is no long-range magnetic order.

In fact, the observation of a large MR without the long-range magnetic order and at temperatures orders of magnitudes higher than the Zeeman energy is already extraordinary. This, together with the fact that it is a semiconducting/insulating system and the MR is positive, is quite surprising because it is in clear contrast with other large MR mechanisms, such as GMR and CMR, where MR is negative and the system has to be metallic or become metallic under magnetic field. This led to our study here that found the rich physics underlying the anomalous MR: the strong coupling between the particle-hole pairing strength and the longitudinal spin susceptibility in the Mott-Slater crossover regime.

Reviewer #3 (Remarks to the Author):

The authors have improved the paper according to my comment and suggestions, and all the issues from me are properly addressed. I also notice that the 2nd reviewer has some concern, and I think it is also important for the authors to further clarify some issues and convince the 2nd reviewer in this round. So I would like to see the comment from the 2nd reviewer in the next round and make the final decision.

Response:

We thank the referee for all the efforts in reviewing our work. We highly appreciate all the constructive and encouraging comments. Please refer to our response to the 2nd referee. Since both comments by referee#2 are domain-related, we point out the fact that the domain effects are not necessary nor sufficient in capturing the physics in the critical regime of the phase transition or above, because they are negligible or nonexistent. We hope this will relieve the remaining concerns.

Reviewer #4 (Remarks to the Author):

The manuscript has been gone through a first-round reviewing by three very competent referees. The reply by the authors was done adequately and professionally. A positive magnetoresistance observed (mainly) in the vicinity of magnetic transition temperature on the artificial superlattice insulator is interesting. The physics proposed behind the positive MR could have an impact in the field of strongly correlated electrons with strong spin-orbit coupling. The data quality in the measurements with a suite of probes is good. The manuscript could be eventually acceptable by Nature Communications; but it must go through a major overhaul; perhaps by adding additional experimental data.

Response:

We thank the referee for the careful review on our work and appreciate for the recommendation. Next, we list our reply to each point.

1) *The concept of Slater-Mott crossover is not clear and may even be misleading. A Mott transition refers a metal-insulator transition due to electron correlations; the magnetic transition which occurs at a low temperature (in some cases) does not influence transport properties dramatically. The Slater transition on the other occurs in a band electron system where a change of the translational symmetry at $T < T_N$ introduces a gap at the Brillouin zone boundary. However, I do agree that a maximum T_N should occur at crossover of localized to itinerant electronic behavior as shown in Fig.4 (a). Fig.1 (abc) is not helpful at all in the introduction for understanding the work carried out in this manuscript and it should be removed. “Spin ordering is to order the hole-electron pair” is simply wrong.*

Response:

We would like to first clarify that “Mott” and “Slater” here in “Slater-Mott crossover” refer to Mott insulator and Slater insulator, respectively, rather than Mott transition and Slater transition. At a given finite temperature, a Hubbard model leads either to the Mott picture under a strong U/t or the Slater picture in case of a weak U/t . At zero temperature, both the Mott and Slater pictures have the AFM insulating ground state with the same symmetry that can be adiabatically

connected by varying U/t [*Phys. Rev. Lett.* 101, 186403 (2008); *Phys. Rev. B* **95**, 235109 (2017); *Phys. Rev. B* **98**, 205114 (2018)]. This is the reason that it is a crossover between the two ends rather than a transition. We now explicitly explained the crossover regime on Page 4 of the manuscript (please also refer to the response#7).

The fact that it is a crossover as a function of U/t often creates controversy because an experimentally observed AFM insulating ground state can be interpreted as a Mott insulator or a Slater insulator. The Iridates are a good example as discussed in the manuscript. The referee is correct that the transport properties should become metallic/itinerant above T_N for Slater insulator and should be insensitive to the AFM transition for Mott insulator, which is well in line with our discussion in the manuscript that “*the disordered state above the phase transition*” holds the key. The issue is that this distinct paramagnetic state behavior between the two limits is no longer distinct in the crossover regime. In other words, although the AFM insulating ground state remains preserved, the high-temperature physical properties cannot be simply described with a Mott picture or a Slater picture due to $U/t \sim 1$ in this regime, which is why the concept of Slater-Mott crossover is crucial.

The characteristic behavior in the crossover regime should be that the charge degrees of freedom are still gapped (insulating) due to formation of particle-hole pairs that are responsible for the fluctuating magnetic moments. The natural consequence is that a charge excitation into the particle-hole continuum will effectively increase the carrier density at the expense of reducing the magnitude of the magnetic moment. This is the message that we tried to convey with Fig.1a-c, because this longitudinal fluctuation is in stark contrast to the usually known transverse motions of spin. To avoid confusion, we have revised it to “Both of these two perturbative approaches predict the same ground state, but neither of them provides a complete description of the experimentally observed behaviors¹⁴. For instance, the charge gap is often found to be reduced from the Mott limit and of a similar size to the magnon bandwidth¹⁵. Meanwhile, unlike the Slater limit, the insulating behavior and the magnetic moments persist above T_N ¹⁶⁻¹⁸. These characters indicate that these materials belong to the crossover regime between the Mott and Slater limits, where neither the Coulomb potential nor the kinetic energy dominates, allowing charge fluctuations to significantly reduce the longitudinal spin stiffness.” on Page 3.

We do not find in our manuscript or supplement the sentence of “*Spin ordering is to order the hole-electron pair*” quoted by the referee. To avoid potential confusion, we rewrote the sentence “*One of the profound outcomes ... is strongly suppressed.*” as “One of the profound outcomes of the electronic spin-charge interplay is the Mott insulating state at half-filling³, where charge localization gives rise to local magnetic moments. The local magnetic moments are thus effectively local particle-hole pairs, and they interact antiferromagnetically with their neighbors and order below the Néel temperature T_N (Fig. 1a). Correspondingly, fluctuations that excite localized charges into the electron-hole continuum above the Mott gap would lead to spatial fluctuations in the size of the magnetic moments, and vice versa (Fig. 1b). It is however difficult to detect and exploit this interplay between spin and charge fluctuations because the charge degree of freedom is often frozen in practical Mott materials, like the parent compounds of high- T_c cuprates⁴, which are often deep inside the Mott regime. Moreover, the local moments are shielded from the external magnetic field by the antiferromagnetic (AFM) interaction.”

2) *The submission package lacks a thorough description of the sample preparation and characterization, for example, the monitoring of 1:1 SIO/STO layers along the c axis. How many layers of the superlattice were used in the sample for the MR measurement?*

Response:

The sample preparation condition and detailed structural characterizations can be found in our previous publication (ref. 23 and the supplement therein). We now added the superlattice thickness in the method section.

3) *In the superlattice, the Ir-Ir separation along the c axis is as larger as 8 Å, whereas it is about 7 Å in Sr₂IrO₄. Perhaps, this difference can be used to justify a lower T_N in the superlattice. The key point is that the canted spin points to the + -b axis alternately along the c axis, so that the net moment vanishes in Sr₂IrO₄. In contrast, they are on the same direction in the superlattice. The magnetization measurements along a,b,c are helpful to establish the magnetic structure (given that there is no neutron diffraction to confirm the magnetic structure). What is the coercive force of this canted iridate superlattice? It is normally pretty large. In case it falls into*

the field range of a PPMS, the MR measurement in a loop of scanning magnetic field could be helpful to answer the question of domain boundary scattering by one of the previous reviewers.

Response:

We appreciate the referee for acknowledging the much simpler crystal and magnetic structure of the superlattice than Sr_2IrO_4 .

The magnetic structure of the superlattice has been unambiguously determined by magnetization measurements and x-ray resonant magnetic scattering. In short, the observation of the (0.5, 0.5, *integer*) magnetic peaks (Fig. 1f) within the supercell confirms the Ir moments are ordered antiferromagnetically within the *ab*-plane and order ferromagnetically along the *c*-axis with the neighboring Ir layer. The much larger in-plane net magnetization than the out-of-plane one indicates an easy-plane anisotropy of the ordered moments with spin canting primarily within the *ab*-plane (Fig. S2 in the supplementary material). More detailed discussions on the magnetic structure can also be found in ref.23.

Following the referee's suggestion, we performed magnetic field scan of the MR at three representative temperatures. In Fig. R1, the coercive field is about 1.5 T at 100 K. The coercive field drops quickly with increasing temperature and the MR is symmetric with no observable hysteresis at temperatures around T_N . This result is consistent with our reply to referee#2 and the previous reply-to-referee letter that the magnetic domain related effects are negligible at temperatures near and above T_N , which is the focused temperature regime in this study.

Figure R1. In-plane MR as a function of magnetic field at different temperatures. The field scanning direction is indicated by an arrow.

4) *What is the resistivity of the superlattice? Do the activated resistance in Fig.1(e) and magnetic property depend on the number of layers? Is it possible to measure the MR on a sample with single layer SIO.*

Response:

The room-temperature resistivity is at the level of 0.01 Ohm*cm (Fig. 2(a) in ref. 22 and Fig. 2(a) in ref. 23). We did not perform a systematic investigation on the thickness dependence of the physical properties, but we have grown another superlattice with 30 repeats in response to the referee's comment. As shown in Fig. R2(a), the resistivity of the thinner superlattice clearly deviates from the thermal activated behavior with a constant activation energy near T_N , consistent with the observation in Fig. 1(e). Additionally, x-ray resonant magnetic scattering measurement shown in Fig. R2(b) demonstrates that the AFM ground state of this superlattice is the same as that shown in Fig.1(f) with very similar T_N . In summary, this thinner sample shows all the characteristic behaviors discussed in the manuscript.

It would certainly be an interesting direction for future investigation to systematically reduce the thickness toward a film of a single SrIrO₃ monolayer. On the other hand, we would like to point

out that the superlattice is essentially a crystal of $\text{Sr}_2\text{IrTiO}_6$, and this study focuses on the bulk properties of this layered system. We now added “The SL structure is effectively an artificial crystal of $\text{Sr}_2\text{IrTiO}_6$ with a confined square lattice of corner-sharing IrO_6 octahedra in the unit cell^{22,23}.” in the manuscript to clarify this point. Varying the number of repeats of the superlattice would effectively be a thickness variation of a $\text{Sr}_2\text{IrTiO}_6$ thin film. It is well known that complex oxides often exhibit a critical thickness below which the bulk properties, especially long-range orders, are suppressed [*Science* 305, 646 (2004)]. This is often referred as “dead layer” in literature with controversial debates on the underlying mechanism, both intrinsic and extrinsic. Therefore, it probably would not be surprising to see significant thickness dependence when a $\text{Sr}_2\text{IrTiO}_6$ thin film approaches a couple of unit cells. Nevertheless, we thank the referee for this suggestion.

Figure R2. (a) Temperature dependent resistivity of a superlattice with 30 repeats. The dashed line denotes the extrapolated thermal activated behavior by fitting the data from 200 K to 300 K. (b) Temperature dependence of the (0.5 0.5 1) magnetic peak intensity around the Ir L_3 -edge.

5) *Ferromagnetic Mott insulator (very few) would show a large negative MR like in the underdoped $\text{La}_{1-x}\text{CaMnO}_3$. On the other hand, metals show a positive MR by following the Kohler's rule. If the resistivity in the superlattice is low (perhaps a good conductor at $T > T_N$), the physics of MR may be simply as in the Kohler's rule. As a matter of fact, the $\text{MR}(H)$ at 175 K in Fig.2 (c) can be well described by the Kohler's rule. Why do the authors fit the $\text{MR}(H)$ in this way and find if it makes sense.*

Response:

We thank the referee for raising this point. Ferromagnetism in Mott insulating manganite, such as LaMnO_3 , requires orbital order. They cannot be described by an effective single-band Hamiltonian. The large negative MR of manganites, often referred as CMR, occurs essentially by turning the insulating state into a metal. This is exactly one of the reasons a large positive MR in an insulating system is rare as discussed in the manuscript.

Regarding the Kohler's rule, as shown in Fig. 1(e), the SL displays an insulating behavior above T_N . Therefore, the Kohler's rule, that usually explains MR phenomena in good metals, is not expected to hold in our sample. This can be further elucidated by showing the Kohler's plot in Fig. R3, where a simple scaling relation of $\Delta R/R_0$ and B/R_0 between different temperatures (175K and 195K) clearly failed, in contrast to the prediction of the Kohler's rule. In fact, the anomalous temperature dependence of the MR indicates a dominant role of spin fluctuations.

Figure R3. Kohler's plot of the SL at two representative temperatures.

6) *In the vicinity of T_N , a large MR effect has been interpreted by suppressing the spin longitudinal fluctuations under magnetic field. Since a dramatic increase of resistance as temperature decreases through T_N , the superlattice looks like a Slater insulator. Can the authors derive a gap (by using the $R(T)$ at lower temperatures and separate the two contributions of*

activated $R(T)$ due to the gap and the extra resistance due to the spin fluctuations in the resistivity change near T_N ?

Response:

According to the referee's suggestion, he/she would like us to break down the resistivity increase near T_N into two components, one from the gap opening due to AFM order and one from spin fluctuations. First of all, we would like to clarify that the spin fluctuations here decrease resistance rather than increasing it. This is the reason why the MR is positive when the field suppresses the spin fluctuations. Secondly, the gap opening in a Slater insulator is exactly the same process as reducing fluctuations because this charge gap is magnetically driven and proportional to the AFM order parameter [*Phys. Rev. B* **95**, 235109 (2017)]. The gap size is strongly temperature dependent and it is significantly reduced near T_N from the lower-temperature value precisely because of thermally enhanced fluctuations. Therefore, assuming a lower-temperature gap size near T_N is in our view somewhat artificial.

7) Fig.1 (d) shows the magnetic structure schematically, whereas the spin structure in Fig.3 is essentially ferromagnetic. Actually, Fig. 3 is not meaningful for understanding the results presented. The authors have introduced key concepts that scattering over places through the text. A clear picture of physics behind the results, such as why the crossover is needed for providing a channel connecting spin and the transport, why strong SOC is needed to give rise a canting (perhaps through the particular superlattice structure) and so on, should be given in the abstract or the introduction.

Response:

We would like to clarify that Fig. 3 illustrates the spin-dependent hopping rather than the magnetic ordering. Since the charge maintains the pseudospin state during hopping, all the arrows point to the same direction in Fig. 3. Due to the important role of the spin dependent hopping, we think it is necessary to illustrate them in addition to writing down the Hamiltonian. For instance, although the DM interaction in the Mott limit are known to be driven by this hopping process [*Phys. Rev.* **120**, 91 (1960)], this superexchange-type approach projects out the charge degree of freedom and is unable to describe electronic behavior. This is why one has to

start from the Hubbard model, where the spin-dependent charge hopping is explicitly included. This in fact helps to clarify the difference between the canting angle and the phase angle and illustrate the staggered field effect in the last reply-to-referee. To avoid confusion, we add both spin-up and spin-down channels in Fig. 3 for a complete representation.

We also added “These characters indicate that these materials belong to the crossover regime between the Mott and Slater limits, where neither the Coulomb potential nor the kinetic energy dominates, allowing charge fluctuations to significantly reduce the longitudinal spin stiffness. The Slater-Mott crossover regime is in fact the particle-hole counterpart of the famous BCS-to-Bose-Einstein condensation (BEC) crossover observed in ultracold-superfluids¹⁹⁻²¹. One may thus anticipate strong spin-charge fluctuations above T_N , that are absent in conventional Mott materials and must be considered by including both spin and charge degrees of freedom in the model Hamiltonian.” in the introduction to explain the Slater-Mott crossover in more detail.

8) *Moreover, more references about SIO are necessary to cover the knowledge base about this material, for example the layer-dependence of transport behavior (PRL 119, 256403) and to show that the perovskite SIO is close to the magnetic instability.*

Response:

Following the referee’s suggestion, we now added the recommended work as ref. 25.

Reviewer #5 (Remarks to the Author):

This paper reports on the coupling between staggered magnetic moment and external magnetic field in a $\text{SrIrO}_3/\text{SrTiO}_3$ superlattice, which shows positive in-plane magnetoresistance. Combined with model calculation, the authors revealed that the positive magnetoresistance originates from the phase factor in the hopping term given by SOI. Apart from detailed preciseness of the descriptions, two main factors are central debates whether this manuscript deserves Nature Communications, which are positive in-plane magnetoresistance near T_N in an insulating regime and the mechanism of the positive magnetoresistance. In my opinion, the MR and the mechanism found by the authors may be new. However, I am not quite positive to

recommend this paper for Nature Communications, given that Nature Communications requires “novelty”. In my opinion, a proper understanding of MR may be scientifically important but just proposing a new mechanism is not very “novel” unless it is very dramatic (e.g. huge MR or in an extremely small magnetic scale) or it gives important physical parameters (e.g. strength of SOI or scattering times in the case of (anti)-weak localization). Nowadays, there are many mechanisms to give various MRs in various conducting regimes from the metallic to hopping conduction. For example, positive magnetoresistance in a similar magnitude is also proposed for the hopping regime [L. Essaleh et al., Phys. Rev. B 52, 7798 (1995); W. Jiang et al., Phys. Rev. B 49, 690 (1994)] although the mechanism may not be the same as the present case. Having said this, the manuscript itself is relatively sound (except the concerns shown below) and should be published somewhere. At least, this paper proposes a possible new mechanism of positive MR involving SOI in the hopping term. However, this manuscript may not be suitable for Nature Communications in my opinion.

Response:

We thank the referee for reviewing our work, and we appreciate for the useful comments and references. In the following, we address this general comment on novelty in three main points.

While very large effects will be important for a fraction of the Nat. Comm. audience focused on applications, many readers will be very interested in the scientific importance of a new mechanism. For example, the first report by P. Grünberg et al. on giant magnetoresistance (GMR) showed an effect of only a few percent [*Phys. Rev. B* 39, 4828(R) (1989)], while the application of GMR devices are well-recognized today in our daily life. Another good example in more recent time is the spin Hall magnetoresistance (<0.01%) mentioned by referee#3 [*Phys. Rev. Lett.* **110**, 206601 (2013)] (citations > 560). From a broader perspective, the discovery of superconductivity in twisted graphene has generated huge interest though it only occurs below 1.7 K [*Nature* **556**, 43 (2018)]. We note that the criteria for Nat. Comm. taken from their website are: 1) *The results are novel (we do not consider abstracts and internet preprints to compromise novelty)*; 2) *The manuscript is important to scientists in the specific field*. Our reading of these criteria is that they suggest the result needs to be scientifically important and not preceded by other substantially similar work. We feel strongly that we meet these criteria.

We certainly agree with the referee that there are many mechanisms for MRs. What makes the mechanism here distinct is that the MR originates from the longitudinal fluctuations, whereas the other mechanisms for magnetic materials rely on the transverse motions of the ordered moments. As discussed in the manuscript, this has an important implication that the associated magnetoelectronic effects can occur and function at much higher temperatures, whereas the mechanisms based on controlling the spin orientation require well-ordered moments below the transition temperature. Fundamentally, this difference is due to the fact that one usually needs to fight against thermal fluctuations for a sizable effect, while the fluctuations here actually help amplify the response to the external field. The MR mechanisms with variable-range hopping discussed by the two references from the referee can be put in the same context. The effect therein becomes bigger at lower temperature because the activation energy ε enters as a factor of $\exp(\varepsilon/k_B T)$, preferring small thermal energy.

The second reference on $Y_{1-x}Pr_xBa_2Cu_3O_7$ is actually a very good example for comparison. The antiferromagnetic parent phase of cuprates can be considered as a Hubbard system but with much larger onsite Coulomb repulsion. Therefore, longitudinal spin fluctuations are strongly suppressed, and indeed no positive anomalous MR was observed near T_N . Our observations and simulations can thus fill the crucial gap in studying the long-standing problem of 2D Hubbard model with an intermediate coupling strength. In addition, we showed how one can design and create a material with such properties (for more details, please see the previous reply-to-referees). This is another important novel contribution of our investigation to the correlated electron community.

Other than the above concern, the following points should be considered.

1. The structure of the sample is still not clear. First of all, the number of periods is not shown. Second, the authors should show experimental data of the sample, for example, by XRD or TEM. (SrIrO₃)₁/(SrTiO₃)₁ superlattice is not a trivial structure to fabricate and experimental evidence is necessary to show the sample used in this study is really (SrIrO₃)₁/(SrTiO₃)₁ superlattice.

Response:

The number of periods is 60. We have added this information in the method section. For the detailed growth parameters, structural characterizations and demonstration of the reproducibility, please see our previous publication (ref. 23) and supplement therein.

2. As pointed out by the other referee, the magnitude of MR (1%/T) is not large, even small. To advertise the magnitude of MR, the authors compared the activation energy and Zeeman energy. However, it is quite rare that Zeeman energy directly couples with the activation energy of electrical conduction. In my opinion, this comparison and the response coefficient of ~6000 is meaningless.

Response:

The referee raised an interesting point. For this system, the Zeeman energy *does* couple with the activation energy, or more precisely the charge gap. As discussed in the manuscript, this unique feature is due to the fact that the conduction carriers and the magnetic moments originate from the same group of spin-orbit-entangled electrons in a Hubbard system/model. Specifically, binding and disassociating a local electron-hole pair correspond to creating and annihilating a magnetic moment, respectively, and vice versa. The external field modulates this energy barrier by coupling linearly to the staggered moments, thanks to spin-orbit interaction. This can be clearly seen in the Hamiltonian, where the Zeeman term in Eq. 1 leads to an effective staggered Zeeman term in Eq. 2 after the staggered transformation. It is indeed rare and has not been exploited before until our study.

Because of the nature of this coupling and referee#3's suggestion on extracting the corresponding parameter from the experimental data, we compared the modulation of the activation energy from MR and the Zeeman energy from XMCD. The former is the charge response, while the latter is the input from the magnetic field. Their ratio is the response coefficient that characterizes the underlying spin-charge interplay in this system. For this reason, we cannot agree with the assessment that this is meaningless. In fact, this is a key "*physical parameter*", as referred by the referee in the general comment, which this MR mechanism can provide. It is the manifestation of the spin susceptibility but in the charge degree of freedom.

Next, we list all other major changes which were not mentioned in the above responses.

Text

1) Line 49.

Delete “*One of the profound outcomes of the electronic spin-charge interplay is the Mott insulating state at half-filling³, where charge localization gives rise to local magnetic moments that order antiferromagnetically below the Néel temperature T_N (Fig. 1a). The local magnetic moments arise from the formation of particle-hole pairs, while the antiferromagnetic (AFM) ordering corresponds to a condensation of these pairs. Correspondingly, the suppression of the AFM moments can be associated with a reduction in the number of particle-hole pairs. This process leads to an increase in the number of free particles and holes promoted to electron-hole continuum above the Mott gap (Fig. 1b). It is however difficult to detect and exploit these spin-charge fluctuations because AFM order is hard to control. Moreover, practical Mott materials, like the parent compounds of high- T_c cuprates⁴, are often deep inside the Mott regime where the charge degree of freedom is frozen and charge transport is strongly suppressed*”.

2) Line 66.

Delete “*The fact that none of these regimes provide a complete description of the experimentally observed behaviors indicates that these materials belong to the intermediate-coupling regime, where neither the Coulomb potential nor the kinetic energy dominates. The resulting reduction of the longitudinal spin stiffness and the charge gap enables strong spin-charge fluctuations, which are absent in conventional Mott materials.*”

3) Line 97.

Delete “*which can be well described within the Mott-Heisenberg scheme*”.

4) Line 193.

Add “above the charge gap”.

5) Line 242.

Add “where magnetic domains are absent”.

6) Figure 3

Add the spin-down channel.

Figure caption

7) Figure 2

Replace “*Charge hopping (spin-up channel) in different spin frames*” with “Charge hopping in spin-up (red) and spin-down channels (blue) in different spin frames”. Delete “*Charge hopping in the spin-down channel can be obtained after applying the time-reversal symmetry operation*”.

8) Figure 4

Replace “*T-dependent resistivity (b), M^{st} (c), MR (circles) and field-induced M^{st} -variation (diamonds) (d), n (e) and field-induced n -variation (f) calculated at zero field (black circles), in-plane field (red up triangles) and out-of-plane field (blue down triangles) for $h \approx 0.1t$ and $U = 3t$.* ” with “*T-dependent resistivity (b), M^{st} (c) and n (e) calculated at zero field (black circles), in-plane field (red up triangles) and out-of-plane field (blue down triangles) for $h \approx 0.1t$ and $U = 3t$. T-dependent MR (circles) and field-induced M^{st} -variation (diamonds) (d), and field-induced n -variation (f) under an in-plane (full) and out-of-plane (open) field.*”.

References

9) Add references:

15. Kim, J. *et al.* Large Spin-Wave Energy Gap in the Bilayer Iridate Sr₃Ir₂O₇: Evidence for Enhanced Dipolar Interactions Near the Mott Metal-Insulator Transition. *Phys. Rev. Lett.* **109**, 157402 (2012).
16. Moser, S. *et al.* The electronic structure of the high-symmetry perovskite iridate Ba₂IrO₄. *New J. Phys.* **16**, 013008 (2014).
17. Zocco, D. A. *et al.* Persistent non-metallic behavior in Sr₂IrO₄ and Sr₃Ir₂O₇ at high pressures. *J. Phys.: Condens. Matter* **26**, 255603 (2014).
18. Fujiyama, S. *et al.* Two-Dimensional Heisenberg Behavior of $J_{\text{eff}} = 1/2$ Isospins in the Paramagnetic State of the Spin-Orbital Mott Insulator Sr₂IrO₄. *Phys. Rev. Lett.* **108**, 247212 (2012).
25. Groenendijk, D. J. *et al.* Spin-Orbit Semimetal SrIrO₃ in the Two-Dimensional Limit. *Phys. Rev. Lett.* **119**, 256403 (2017).
26. Schütz, P. *et al.* Dimensionality-Driven Metal-Insulator Transition in Spin-Orbit-Coupled SrIrO₃. *Phys. Rev. Lett.* **119**, 256404 (2017).
35. Brandow, B. H. Electronic structure of Mott insulators. *Adv. Phys.* **26**, 651-808 (1977).

All the in-text citations are also updated.

Reviewers' Comments:

Reviewer #2:

Remarks to the Author:

The authors have responded to this reviewer's comments with some reasoning, but their rebuttal has not clearly answered the raised comments.

1) The author's claim may be valid, if the assumption that this magnetoresistance-component subsisting well below T_N (authors call as "the extrinsic effect") disappears near T_N is verified. As pointed out previously, however, the model calculation only explains the peak-like temperature dependence nearby T_N and does not provide clear reason why the magnetoresistance-component subsisting well below T_N can be ignored nearby T_N . Thus, I think that the author's discussion remains to be insufficient.

2) I still feel that the result appears to be ordinary, which may not reach the criteria of Nature Communications.

Reviewer #3:

Remarks to the Author:

After reading the comments from the other reviewers, especially the 2nd reviewer, my feeling is that the revised manuscript has been improved substantially. It is true that the 2nd and 5th reviewers have some concern about the novelty of the presented result, but for this reviewer, I think the present manuscript can be published in Nature Communications as it is. I agree that the explanation still has some weakness as the other reviewers point out, but it is also not reasonable to ask the authors to thoroughly understand all the details in one paper. Personally, I feel inspired by the observations and it fits my standard for a Nature Communications paper.

Reviewer #4:

Remarks to the Author:

The authors addressed questions adequately by a carefully prepared reply, modifying the text and performed additional experiments. Their finding and explanation would be interesting to the community. Therefore, I recommend to accept the paper without further delay.

Reviewer #5:

Remarks to the Author:

I read through the response from the authors and find that most of my previous concerns are responded satisfactorily. However, I am still not well convinced of "novelty" of the manuscript proposing a new mechanism of the positive MR. As in the case of GMR, it is difficult to judge whether a proposed mechanism of MR contains substantial scientific importance or will lead to some applications at the stage of a first report. The significance of such a proposal will usually be clear after some amount of related studies are carried out. In this sense, this study would give a big impact but, at the same time, I know many papers proposing various mechanisms of MR which are rarely recognized broadly due to some reasons such as lack of generality. I think, while this paper should be published somewhere, the manuscript does not satisfy the criterion of the novelty of Nature Communications at the present stage.

Reviewer #2 (Remarks to the Author):

The authors have responded to this reviewer's comments with some reasoning, but their rebuttal has not clearly answered the raised comments.

Response:

We thank the reviewer for reviewing our work, below we address the remaining concerns.

1) The author's claim may be valid, if the assumption that this magnetoresistance-component subsisting well below T_N (authors call as "the extrinsic effect") disappears near T_N is verified. As pointed out previously, however, the model calculation only explains the peak-like temperature dependence nearby T_N and does not provide clear reason why the magnetoresistance-component subsisting well below T_N can be ignored nearby T_N . Thus, I think that the author's discussion remains to be insufficient.

Response:

Contributions from magnetic domain reversal to magnetoresistance are inevitable at temperatures well below the magnetic ordering temperature. But domains are weakened as the order parameter decreases with increasing temperature. Their effects are thus the weakest near the transition and finally disappear above the transition. This can be seen in the hysteretic behavior from the data shown in the 1st response to the reviewer#1 (in the first response letter), the 3rd response to the reviewer#4 (in the second response letter) and Fig. S4 in the supplementary material. In other words, the diminishing domain effects with increasing temperature cannot account for the peak in MR near T_N , i.e., the anomalous MR. Therefore, we chose to neglect them in order to focus on the truly dominant effect, which is the magnetic fluctuations and requires a very different approach to simulate, especially at the high temperature region. Our results indeed show that the theoretical model is able to capture the critical fluctuations very well. Although discrepancy between the theoretical model and experimental observation occurs in the low-temperature region, it is as-expected and does not alter the conclusion of this work. Nevertheless, we thank the reviewer for raising this comment. Developing computational methods that could account for different physics at all temperatures would certainly be an interesting direction for future investigations.

2) *I still feel that the result appears to be ordinary, which may not reach the criteria of Nature Communications.*

Response:

As we discussed in the manuscript and the previous replies, our findings offer the community an alternative consideration of the intermediate regime of the Hubbard model from the normal state perspective, and present a model system that can be engineered to exploit the spin-charge fluctuations. Other reviewers share the same view of inspiration from the results. We feel sorry that Reviewer #2 did not reach a similar opinion. Nevertheless, we thank the reviewer for the comments and we feel that our work well meets the publish criteria of *Nature communications*.

Reviewer #3 (Remarks to the Author):

After reading the comments from the other reviewers, especially the 2nd reviewer, my feeling is that the revised manuscript has been improved substantially. It is true that the 2nd and 5th reviewers have some concern about the novelty of the presented result, but for this reviewer, I think the present manuscript can be published in Nature Communications as it is. I agree that the explanation still has some weakness as the other reviewers point out, but it is also not reasonable to ask the authors to thoroughly understand all the details in one paper. Personally, I feel inspired by the observations and it fits my standard for a Nature Communications paper.

Response:

We thank the reviewer's effort in reviewing our work. We are highly encouraged by the reviewer's recognition of the value of our work and appreciate the positive recommendation.

Reviewer #4 (Remarks to the Author):

The authors addressed questions adequately by a carefully prepared reply, modifying the text and performed additional experiments. Their finding and explanation would be interesting to the community. Therefore, I recommend to accept the paper without further delay.

Response:

We thank the reviewer's careful review of our work and value all the constructive criticisms. We highly appreciate for the recommendation for publication on *Nature communications*.

Reviewer #5 (Remarks to the Author):

I read through the response from the authors and find that most of my previous concerns are responded satisfactorily. However, I am still not well convinced of “novelty” of the manuscript proposing a new mechanism of the positive MR. As in the case of GMR, it is difficult to judge whether a proposed mechanism of MR contains substantial scientific importance or will lead to some applications at the stage of a first report. The significance of such a proposal will usually be clear after some amount of related studies are carried out. In this sense, this study would give a big impact but, at the same time, I know many papers proposing various mechanisms of MR which are rarely recognized broadly due to some reasons such as lack of generality. I think, while this paper should be published somewhere, the manuscript does not satisfy the criterion of the novelty of Nature Communications at the present stage.

Response:

We thank the reviewer a lot for reviewing our works and appreciate all the comments. We are pleased to see the previous concerns are now relieved. We understand that the reviewer would like to see the generality of any new mechanism. While it's impossible to fit all future works into one single manuscript, we did launch another experiment to test the proposed mechanism and criteria in the Discussions section. The data is now shown in Fig. S6 in the Supplementary information, where a similar anomalous MR is observed around the magnetic critical temperature and related discussion can be found in the 2nd response to the 1th reviewer in the first response letter. We also agree with the reviewer that it is difficult to judge a new physics concept in its early stage. However, the above statement is not odd with the importance of this work. We believe that this work affords stimulating findings of significant values to the community and is deserved to be published on *Nature communications*.